# Probing T-cell response by sequence-based probabilistic modeling

**Barbara Bravi**[1], **Vinod P. Balachandran**[2], **Benjamin D. Greenbaum**[3], **Aleksandra M. Walczak**[1☯], **Thierry Mora**[1☯], **Rémi Monasson**[1☯], **Simona Cocco**[1☯]*

**1** Laboratoire de Physique de l'Ecole Normale Supérieure, ENS, Université PSL, CNRS, Sorbonne Université, Université de Paris, Paris, France, **2** Immuno-Oncology Service, Human Oncology and Pathogenesis Program, Hepatopancreatobiliary Service, Department of Surgery, Memorial Sloan Kettering Cancer Center, New York, New York State, United States of America, **3** Computational Oncology, Department of Epidemiology and Biostatistics, Memorial Sloan Kettering Cancer Center, New York, New York State, United States of America

☯ These authors contributed equally to this work.
* simona.cocco@phys.ens.fr

**Data Availability Statement:** All relevant data are within the manuscript, its Supporting information files, and the Git repository (https://github.com/bravib/rbm_tcell).

## Abstract

With the increasing ability to use high-throughput next-generation sequencing to quantify the diversity of the human T cell receptor (TCR) repertoire, the ability to use TCR sequences to infer antigen-specificity could greatly aid potential diagnostics and therapeutics. Here, we use a machine-learning approach known as Restricted Boltzmann Machine to develop a sequence-based inference approach to identify antigen-specific TCRs. Our approach combines probabilistic models of TCR sequences with clone abundance information to extract TCR sequence motifs central to an antigen-specific response. We use this model to identify patient personalized TCR motifs that respond to individual tumor and infectious disease antigens, and to accurately discriminate specific from non-specific responses. Furthermore, the hidden structure of the model results in an interpretable representation space where TCRs responding to the same antigen cluster, correctly discriminating the response of TCR to different viral epitopes. The model can be used to identify condition specific responding TCRs. We focus on the examples of TCRs reactive to candidate neoantigens and selected epitopes in experiments of stimulated TCR clone expansion.

## Author summary

Large repertoires of immune cells, such as T cells, are increasingly made available by high-throughput sequencing. Exploiting such datasets to infer how T cells respond to antigens could help design vaccines and adoptive T-cell therapies. We here propose an approach based on probabilistic machine learning to identify and characterize responding T cells. After learning, this approach is able to distinguish clones that specifically respond to different antigen stimulations. The model parameters and the low-dimensional representations of the T-cell sequences identify sequence motifs underlying T-cell recognition at the molecular level. The approach is illustrated on repertoire data describing *in vitro*

**Funding:** A.M.W is the recipient of the European Research Council Consolidator Grant n. 724208 and the European Research Council Marie Curie-Sklodowska ITN QuanTI; R.M. is the recipient of the ANR-17 RBMPro CE30-0021 grant from the Agence Nationale de la Recherche; S.C. is the recipient of the ANR-19 Decrypted CE30-0021-01 grant and the ANR Flash Covid 19 - FRM, PROJET 'SARS-Cov-2immunRNAs' from the Agence Nationale de la Recherche; T.M. is the recipient of the ANR-19-CE45-0018 RESP-REP grant from the Agence Nationale de la Recherche and the European Research Council Marie Curie-Sklodowska ITN QuanTII; B.D.G. is the recipient of the NIH grant U01CA224175, the Memorial Sloan Kettering Cancer Center grant, and the Pershing Square Sohn Prize-Mark Foundation Fellowship supported by funding from the Mark Foundation For Cancer Research; V.P.B. and B.D.G. are the recipients of a collaboration grant by Stand Up To Cancer, a program of the Entertainment Industry Foundation, and the Lustgarten Foundation. B.B. was funded by the Stand Up To Cancer collaboration grant and the European Research Council Consolidator Grant n. 724208. The funders had no role in study design, data collection and analysis, decision to publish, or preparation of the manuscript.

**Competing interests:** I have read the journal's policy and the authors of this manuscript have the following competing interests: B.G. has received honoraria for speaking engagements from Merck, Bristol-Meyers Squibb, and Chugai Pharmaceuticals, has received research funding from Bristol-Meyers Squibb, and has been a compensated consultant for Darwin health, PMV Pharma and Rome Therapeutics of which he is a cofounder.

stimulation of T cells by cancer-related neoantigens, as well as on data for common infectious diseases.

## Introduction

T cell receptors (TCR) are the key factors through which the adaptive immune system controls pathogens and tumors. T-cells recognize infected and malignant cells by binding antigens, short peptides that are presented on the cell surface by the Major Histocompatibility Complex (MHC) molecules. Upon antigen recognition, an activated T cell divides multiple times, giving rise to a population of T cells with the same TCR—an expanded T-cell clone or clonotype. Clonotype counts provide estimates of T-cell clone abundance and measure the antigen-induced clonal expansion characteristic of an immune response. In an immune response, multiple TCRs can recognize the same antigen and clonally expand. Therefore, TCRs from expanded clonotypes can contain specific information about the antigen that led to their expansion.

High-throughput sequencing of TCR repertoires isolated from patient tissue and blood has been motivated by a desire to both understand response to immunotherapies, such as checkpoint blockade inhibitors [1], and to drive emerging approaches such as anti-cancer vaccines [2, 3] and adoptive T cell therapies [4]. However, repertoire sequencing (RepSeq) datasets are rarely accompanied by structural information on the TCR:pMHC complex [5–7]. These fast-growing sequence datasets give hope that machine-learning approaches can extract information on T-cell recognition from TCR sequences alone. This task is particularly challenging in cancer, where antigens derived from tumor-specific mutations, called *neoantigens*, can activate T-cells and drive tumor regression. The majority of neoantigens are unique to the tumor in question, and response by neoantigen-targeting T cells is expected to be extremely individualized (private), and driven by neoantigens generated by a small fraction of the mutations in a tumor [8].

Neoantigens capable of driving an immune response, and their cognate TCRs, are therefore currently highly sought-after targets for the design of personalized immunotherapies [2–4]. Two outstanding, strongly complementary questions are: what clones specifically respond to a given neoantigen and what specific "features", such as the biochemical properties of the amino acids that make up the TCR, are central to a productive response. Computational methods can help answer these questions by narrowing down the number of candidate responding TCRs, which accelerates experimental testing, typically a labor- and resource-intensive task. In particular, methods aimed at learning a sequence representation of T-cell response can shed light on the binding mechanisms determining T-cell response specificity at the molecular level. Such identification of sequence motifs is a preliminary step to finding TCR residues involved in antigen recognition. From this point of view, TCR sequence-based approaches can also complement and improve other existing approaches, that are focused either on predicting antigen "immunogenicity" [9, 10] or on characterizing T-cell expansion [11].

We propose a set of probabilistic-modeling approaches, based on Restricted Boltzmann Machines [12–14] and previously defined selection factors [15], to characterize features of responding TCR from sequence data. We apply the method to T-cell subrepertoires from [16] that are *in vitro* stimulated by patient-specific tumor neoantigens in the peripheral blood of seven long-term survivors of pancreatic ductal adenocarcinoma (PDAC). The computational approaches we developed discriminate between specific and non-specific expansion and establish a connection between antigen specificity and sequence motifs in TCRs. The methods are general and, apart from the T-cell repertoires from [16], we apply them also to TCR repertoires

responding to specific viral epitopes [17]. In the later case, we show that our methods allow for the identification of clusters of TCRs responding to the specific epitopes.

## Results

### Dataset structure

We consider T-cell repertoires from the *in vitro* T-cell assays that were performed by Balachandran et al. [16] to detect lasting circulating T-cell reactivity to patient-specific neoantigens in PDAC long-term survivors. This cohort consists of seven patients, labeled here as Pt1, . . ., Pt7. First, putative neoantigens were selected among all short peptides harboring a point mutation identified in patients' tumors using the neoantigen fitness model proposed by Łuksza et al. [9]. This fitness model predicts the potential of a neoantigen of being immunogenic (*i.e.*, to trigger an immune response) based on: its sequence homology to known infectious disease-derived antigens, which is a proxy for the probability of its recognition by T cells; and its relative MHC-I binding affinity when compared to the wild-type peptide from which it was derived, quantifying its differential degree of presentation compared to the wild-type. The mutated peptide with the maximum fitness within a tumor was predicted as the immunodominant neoantigen. Subsequently peripheral blood mononuclear cells (PBMCs) from each patient were pulsed *in vitro* with the putative immunodominant neoantigen(s) (NA) for that patient. For comparison, PBMCs were also pulsed with the model predicted homologous infectious disease-derived antigens ("cross-reactive" antigens, CR) of the NA and their corresponding unmutated versions ("wild-type" antigens, WT). In addition, neoantigens were also found to be enriched in the genetic locus MUC16. As a result, MUC16-derived neoantigens were also tested for an *in vitro* response in two patients (Pt1 and Pt2). Overall, this dataset provides 23 antigen-stimulated T-cell repertoires, one for each antigen tested across the seven patients, as summarized in Table 1.

In these assays, T-cell response was monitored in terms of TCR clonal expansion 21 days after *in vitro* stimulation by the selected peptides (Fig 1A). Cytokines Interleukin-2 (IL-2) and Interleukin-15 (IL-15) were added on day 2 and every 2–3 days. Peptides were also re-added on day 7 and 14 for second and third rounds of restimulation. On day 21, cells were restimulated in the presence of peptides for 5 hours before being stained. Sequence reads of the complementarity-determining region 3, CDR3 (the region in contact with antigens) of the TCR $\beta$

**Table 1. List of TRB RepSeq datasets from Ref. [16].** Datasets are organized by patient and by antigens used for the *in vitro* stimulation of the patient's PBMC. Neoantigens tested were chosen based on a measure of neoantigen "fitness" that accounts for: the neoantigen binding affinity to the patient's MHC; its potential for being recognized by T cells, quantified by sequence similarity to infectious disease-related epitopes ("cross-reactive"), see [9, 16]. The neoantigens are generated by cancer-specific point mutations in antigens natural to the organism ("wild-type"). The green circle indicates neoantigens from the genetic locus MUC16, which were not selected based on neoantigen fitness.

| Patient (Pt) | Epitopes: NA (Neo-Antigen), WT (Wild-Type), CR (Cross-Reactive) | Cross-reactive origin |
|---|---|---|
| 1 | **NA** (NLLGRNSFK), **WT** (NLLGRNSFE), **CR** (LLGRNSFEV) | Tumor antigen from p53 |
| 1 | • **MUC16** (TTSPSNTLV), **MUC16WT** (TTSPSTTLV) | |
| 2 | **NA** (QEFENIKSY), **WT** (QEFENIKSS), **CR** (QRFHNIRGRW) | Human papillomavirus |
| 2 | • **MUC16** (EASSAVPTV), **MUC16WT** (EASSTVPTV) | |
| 3 | **NA** (RVWDIVPTL), **CR** (KPWDVVPTV) | Dengue virus |
| 4 | **NA** (LLLMSTLGI), **WT** (LSLMSTLGI), **CR** (LLMGTLGIV) | Human papillomavirus |
| 5 | **NA** (QTYQHMWNY), **CR** (AFWAKHMWNF) | Hepatitis C virus |
| 6 | **NA1** (LPRQYWEAL), **CR1** (KLLPEGYWV) | Francisella tularensis |
| 6 | **NA2** (RPQGQRPAL), **CR2** (SPRGSRPSW) | Hepatitis C virus |
| 7 | **NA** (GIICLDYKL), **CR** (TMGVLCLAIL) | Dengue virus |

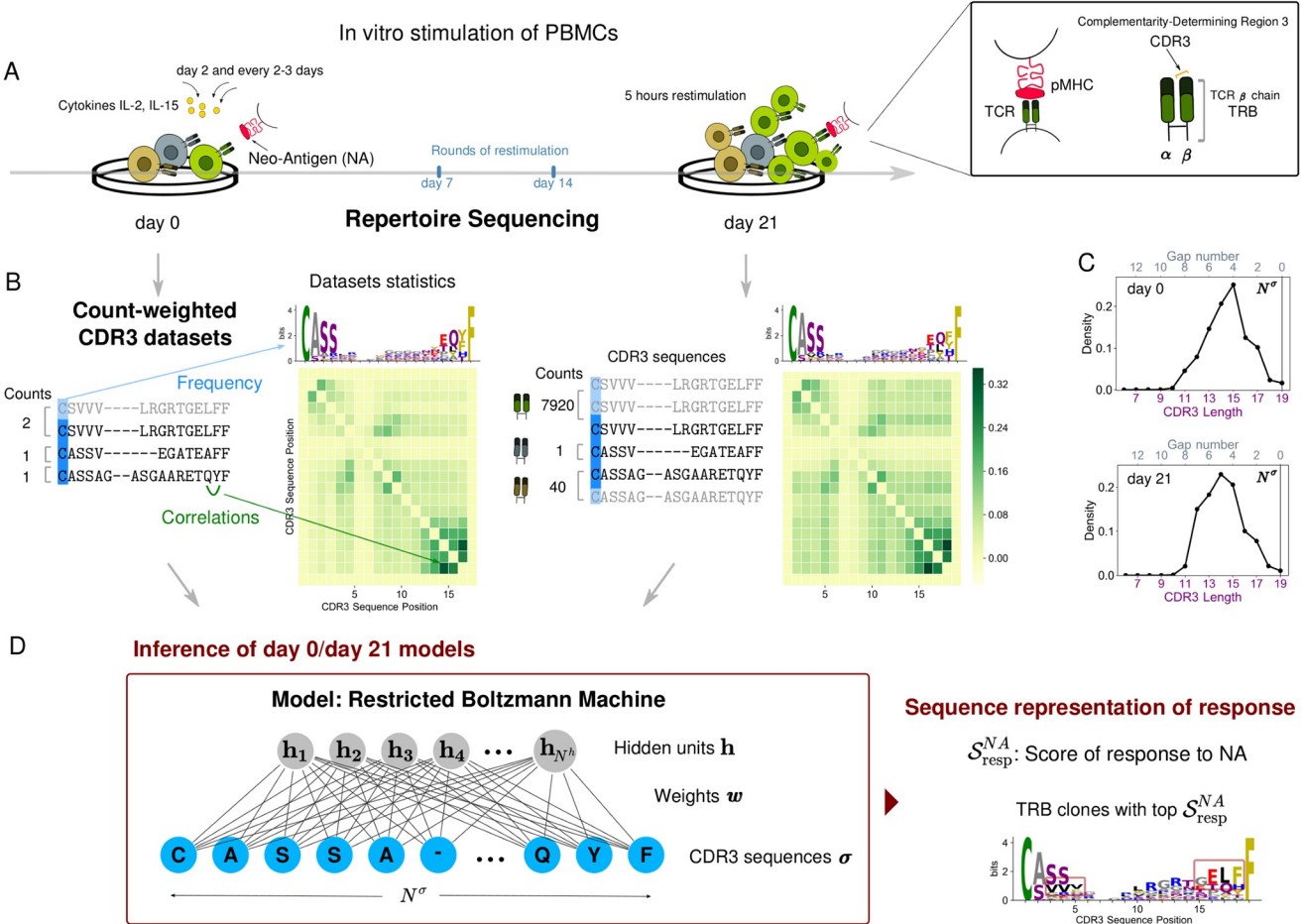

**Fig 1. Schematic of the approach.** A: Response to the antigen occurs via clonal expansion, where responding T cells proliferate by cell reproduction. B: Structure of count-weighted training datasets. Repertoire statistics before (left) and after (right) stimulation can be illustrated in terms of logos that convey information about position dependent residue single-site frequency, and correlation matrices. C: Distribution of CDR3 lengths without alignment (lower $x$-axis). The number of gaps inserted in the sequence center by the alignment (upper $x$-axis) is equal to the difference between the alignment length ($N^\sigma = 19$) and the CDR3 original length. D: The RBM model graphical architecture. The ratio of the RBM model scores before and after stimulation highlights sequence motifs that are characteristic to antigen stimulation (dark red boxes). Data shown for stimulation by NA of patient Pt3 PBMC from Ref [16].

chain (TRB) were obtained both before and after antigen stimulation. We will refer to these lists of sequence reads as the "day 0" and the "day 21" datasets respectively, providing snapshots of the TCR composition of patient repertoires at day 0 (before antigen stimulation) and at day 21 (after *in vitro* stimulation). A clonotype was defined to include all TRBs with identical CDR3 amino acid sequences and we will use this definition. Ref. [16] assessed TRB clonal expansion using flow cytometry and CDR3 sequence reads. They found identical TRB clones that were significantly expanded in response to both NA and CR in all patients and neoantigen-reactive clones that were also present in the same patient's archival primary tumors in 5 out of 7 patients.

Our goal is to learn from CDR3 sequence data sequence-level models capturing information about the response to specific antigens in each T-cell repertoire. We use TRB clonal abundance to train models that identify responding clonotypes. We assign to each CDR3 sequence a multiplicity (counting up the corresponding reads) and we construct sequence datasets where sequences are replicated as many times as given by their multiplicity. In this way, the

contribution of each CDR3 sequence is effectively weighted by counts of sequence reads and we will refer to these training datasets as "count-weighted datasets" (Fig 1B).

## Probabilistic models inferred from sequence data

We learn a distribution of the probability of finding a specific CDR3 sequences in a given repertoire. The statistics describing CDR3 amino acid usage evolve from day 0 to day 21, reflecting the underlying evolution of the T-cell repertoire composition induced in the experiment. TRB clonal expansion modifies the amino acid sequence statistics (Fig 1B). The goal of our inference frameworks is to reconstruct from these changes in the CDR3 sequence statistics the "constraints" (or patterns) at the level of amino acid sequences that characterize antigen induced expansion. We consider three methods: the Restricted Boltzmann Machine (RBM), a selection-factor based method SONIA [15], and an RBM that does not require multiple sequence alignment (RBM-*Left+Right*).

**Restricted Boltzmann machine (RBM).** In the Restricted Boltzmann Machine (RBM) [12, 13] method presented in Fig 1D, the CDR3 sequences (observed units) are coupled to a layer of hidden units by a set of connections called "weights". The advantage of hidden units, when learning a probability distribution of finding a given CDR3 sequence in the repertoire, is that they help capture global correlations between observed units (*i.e.*, along CDR3 sequences), each of the hidden units extracting a feature. The parameters defining the RBM distribution (Eq 2 in Materials and methods): the local biases acting on the observed units and the global weights are inferred from the count-weighted datasets representing each TRB repertoire, in such a way as to reproduce the repertoire statistics shaped by the clonal expansion, see Fig 2A, Materials and methods, and S4 Fig.

CDR3 length can vary from a few up to 30 amino acids (S1 Fig). To learn the RBM, we first need to reduce sample sequences to the same length by performing a CDR3 sequence alignment [18]. We propose for CDR3s a novel alignment procedure where we first build an alignment profile of all the sequences of the same length and then progressively re-align profiles of increasing length to obtain a multiple sequence alignment of length $N^\sigma = 19$ amino acids (see Materials and methods, [19]). For sequences that share strongly conserved anchor residues, as in the case of CDR3, the resulting alignment concentrates gaps, due to variable CDR3 length (Fig 1C), in the middle of the sequence and away from the conserved residues. In Fig 1B gaps are indicated by '−' symbols in the lists of CDR3 sequences but excluded from logos to emphasize amino acid usage.

**SONIA.** We also consider a recent computational tool, SONIA [15] that infers selection factors acting on the TRB chain from RepSeq datasets. SONIA quantifies selection pressures on amino acids in the TCR sequences in terms of position-specific "selection" or $q$-factors, by comparing the probability of finding a given sequence in a post-stimulation repertoire of interest compared to a baseline repertoire. In the context of this paper, we model TRB clone expansion as a form of selection pressure imparted by antigen stimulation and learn two SONIA models for the probability of finding a CDR3 sequence, one before and one after antigen stimulation. Specifically, SONIA fits the statistics of count-weighted CDR3 datasets by an independent-site model, fixing the same "background" distribution for both datasets to the probability of generation of these sequences $P_{\text{gen}}$ (Eq 7 in Materials and methods). We used SONIA models with *Left+Right* feature encoding [15] (Fig 2B). Each CDR3 sequence is encoded by a string of 1 and 0, describing the presence or absence of a given amino acid at a given distance from the left end of the CDR3 (the start CDR3 anchor residue) and from the right end (the end CDR3 anchor residue), up to a maximum distance that we set to $N^\sigma = 19$.

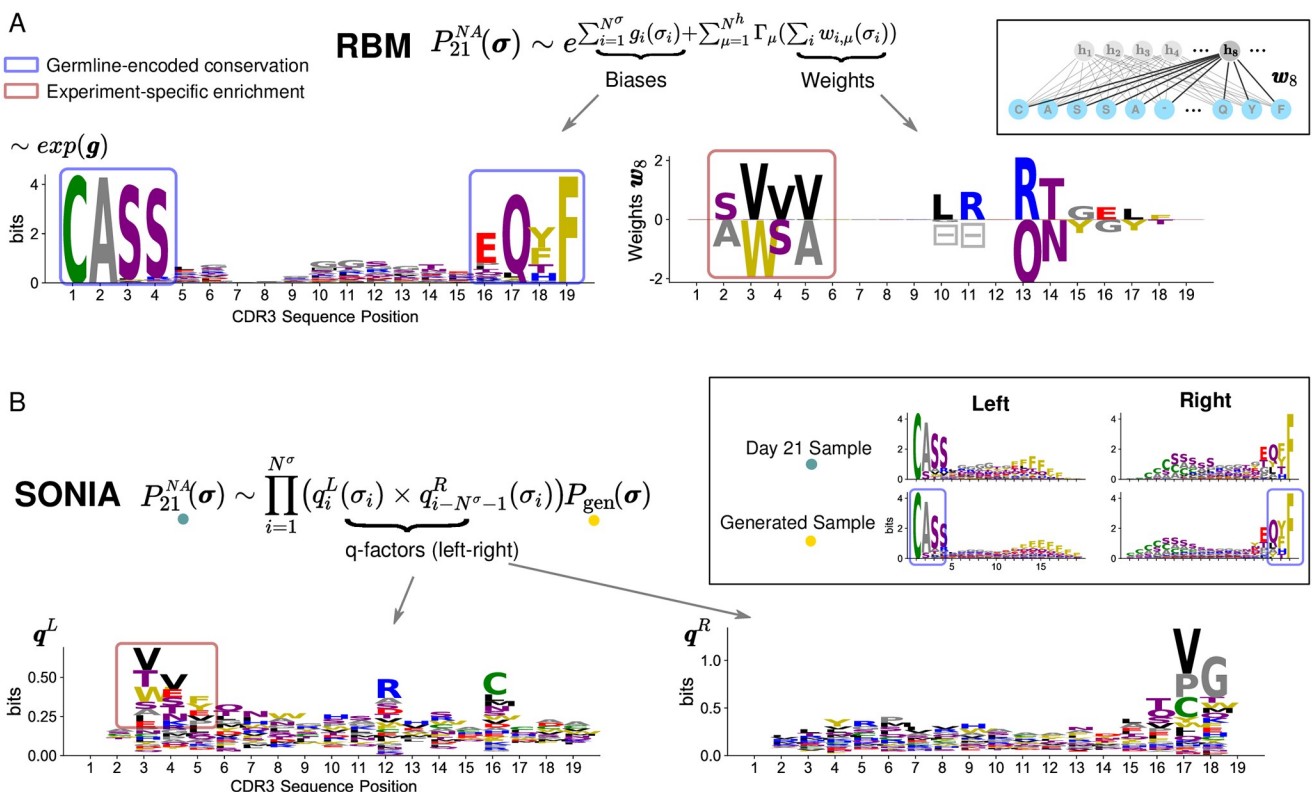

**Fig 2. Inferred model parameters enable feature extraction.** A: Using count-weighted datasets, the RBM infers the probability $P_{21}^{NA}(\boldsymbol{\sigma})$ of detecting a CDR3 sequence $\boldsymbol{\sigma}$ with high abundance at day 21 post stimulation by NA. $P_{21}^{NA}(\boldsymbol{\sigma})$ is parametrized in terms of local biases $\boldsymbol{g}$ acting on CDR3 sites (left column) and weights connecting each CDR3 site to a hidden unit, effectively coupling the CDR3 sites (right column). Inset: example of weights entering hidden unit 8, $\boldsymbol{w}_8$. The logo representation of $\exp(\boldsymbol{g})$ normalized (left column) shows that single-site biases, by capturing the frequency of amino acids at each CDR3 site, reflect mainly the preferential usage of certain residues during the receptor generation process (*e.g.* Cysteine at the beginning of the CDR3, Phenylalanine at the end position, blue boxes). For a given hidden unit, there is a set of weights for each site, with different values for each amino acid appearing at that site. These values can be either positive or negative and the height of the letter reflects the weight's magnitude for the respective amino acid. Symbols ⊟ give weight values for the gap. Weights carry information mainly about experiment-specific enrichment in sequence motifs (*e.g.* VVV or WSA at positions 3–5, dark red box). B: The probability $P_{21}^{NA}(\boldsymbol{\sigma})$ inferred by SONIA is expressed in terms of selection or *q*-factors, one set for the left CDR3 alignment ($\boldsymbol{q}^L$) and one set for the right CDR3 alignment ($\boldsymbol{q}^R$), here represented as sequence logos. Inset: sequence logos at day 21 post-stimulation and for the baseline distribution $P_{\mathrm{gen}}$ with left and right alignment. In contrast to RBM biases, *q*-factors quantify only the enrichment (dark red box) with respect to the TRB generation amino acid usage preferences described by $P_{\mathrm{gen}}$ (blue box in the inset). The gaps appearing in the RBM motifs have been removed from the logo representations in all figures for clarity of presentation. The dataset used is the same as in Fig 1 (stimulation by NA of Pt3 PBMC from [16]).

**RBM-*Left+Right* (RBM-LR).** Finally, we also implemented an RBM version which does not rely on a first step of sequence alignment by performing a *Left+Right* CDR3 encoding as in the SONIA *Left+Right* model [15]. We refer to this RBM version as RBM-LR. The *Left+Right* data encoding is specifically designed to exploit the structure of CDR3 sequences, that are produced by a gene rearrangement process called VDJ recombination [20]: as a result, the start site is a cysteine encoded by the variable (V) segment, the end site is Phenylalanine or Valine encoded by the joining (J) segment and the variability in sequence composition is concentrated in the middle, arising from untemplated insertions and deletions at the junctions with the diversity (D) segment. This left-right encoding has the advantage of not relying on multiple sequence alignments, which are sample- and procedure-dependent, but it is suitable only for biological sequences that display strongly conserved start and end residues, such as CDR3s. The results of the alignment procedure, with gaps located in the central variable region of the CDR3, are consistent with the left-right encoding (see Fig 2 and S2 Fig), proving the ability of our alignment routine to well reflect the known biological structure of CDR3 sequences. We

use in fact the RBM-LR model to show that our conclusions do not depend on data encoding: since the RBM-LR model does not require any alignment, it avoids producing alignment gap artifacts found in the RBM motifs (see S2 Fig and Fig 5), however its training time can be substantially longer (Materials and methods).

## Model parameters are potential tools for motif discovery

The sequence-based approaches in the previous section provide a scoring scheme that associates sequence abundance, which is informative about TRB response, to sequence features, that embed molecular details related to TRB specificity (Fig 1). In particular, the inferred model parameters identify and allow us to derive sequence motifs containing CDR3 residues characteristic of antigen-specific response (Fig 2). The RBM biases $g_i(\sigma_i)$ identify conserved positions along the CDR3 that are mainly a result of preferential usage of these amino acids during the generation process. Similarly SONIA $q$-factors disentangle, in a site-specific way, enrichment in amino acid usage due to clonal expansion from the generated frequency distribution (incorporated in the reference probability $P_{\text{gen}}$). Additionally, RBM weights pick up sequence features coming from global selection patterns that independent-site $q$-factors cannot capture (Fig 2 and S2 Fig). Inferred model parameters can be applied as a motif discovery tool that, once combined with systematic experimental tests, provide specific insights about TCR responses in terms of TCR:pMHC binding.

## Model scores correlate to T-cell clonal expansion

From the dataset incorporating clonal multiplicity upon stimulation by antigen $p$ at day 21, the RBM approach and SONIA learn the probability $P_{21}^p(\boldsymbol{\sigma})$ that a given TRB clone $\boldsymbol{\sigma}$ is detected with high abundance post-stimulation (Fig 2).

As a preliminary step, we asked whether $P_{21}^p(\boldsymbol{\sigma})$ correlates with CDR3 counts at day 21 for sequences in a test set (see Materials and methods). This indicator of performance is extremely poor when we use all the clones, since for low-count clones there is no more correlation between RBM scores and counts (S3(C) Fig). If we filter out low-count clones, progressively restricting the test set to the clones with highest abundance, we recover a significant correlation and RBM performance is generally better than the SONIA biophysical model based on the independent-site assumption (S3 and S4B and S4E Figs). We next introduced a probabilistic score of response for a clone $\boldsymbol{\sigma}$ to stimulation by antigen $p$, $\mathcal{S}_{\text{resp}}^p(\boldsymbol{\sigma}) = \log\left(P_{21}^p(\boldsymbol{\sigma})/P_0(\boldsymbol{\sigma})\right)$, which measures the ratio of the model probability assigned to sequence $\boldsymbol{\sigma}$ in the $p$-specific repertoire after stimulation, $P_{21}^p(\boldsymbol{\sigma})$, relative to the one before stimulation $P_0(\boldsymbol{\sigma})$. We tested the response score's ability to correlate to actual clonal expansion under the same stimulation $p$, quantified by the clone's fold-change with respect to its abundance at day 0 (see Eq 1 in Materials and methods). Fig 3 shows good performance, with correlation coefficients ranging between 0.68-0.83 (RBM) and 0.51-0.71 (SONIA), the only exceptions being visible in datasets Pt7 and Pt4 WT. Here, relatively high counts at day 0 obstruct information about the antigen-specific response at day 21. The model can therefore be used to identify both experimental and condition specific responding TCRs, a task of great importance to advancing personalized immunotherapies. Response scores $\mathcal{S}_{\text{resp}}^p(\boldsymbol{\sigma})$ can also be defined in terms of the TCR probability at day 21 relative to baseline distribution $P_{\text{gen}}(\boldsymbol{\sigma})$ [21]. These scores capture information about the overall selection pressures acting on the sequence, including thymic and antigen specific selection [15] (see Materials and methods). The advantage of this choice is that $P_{\text{gen}}$ can be estimated *in silico* [21], without the need for longitudinal datasets that include sequence data at time points before stimulation, which are not always available. More generally, $\mathcal{S}_{\text{resp}}^p(\boldsymbol{\sigma})$

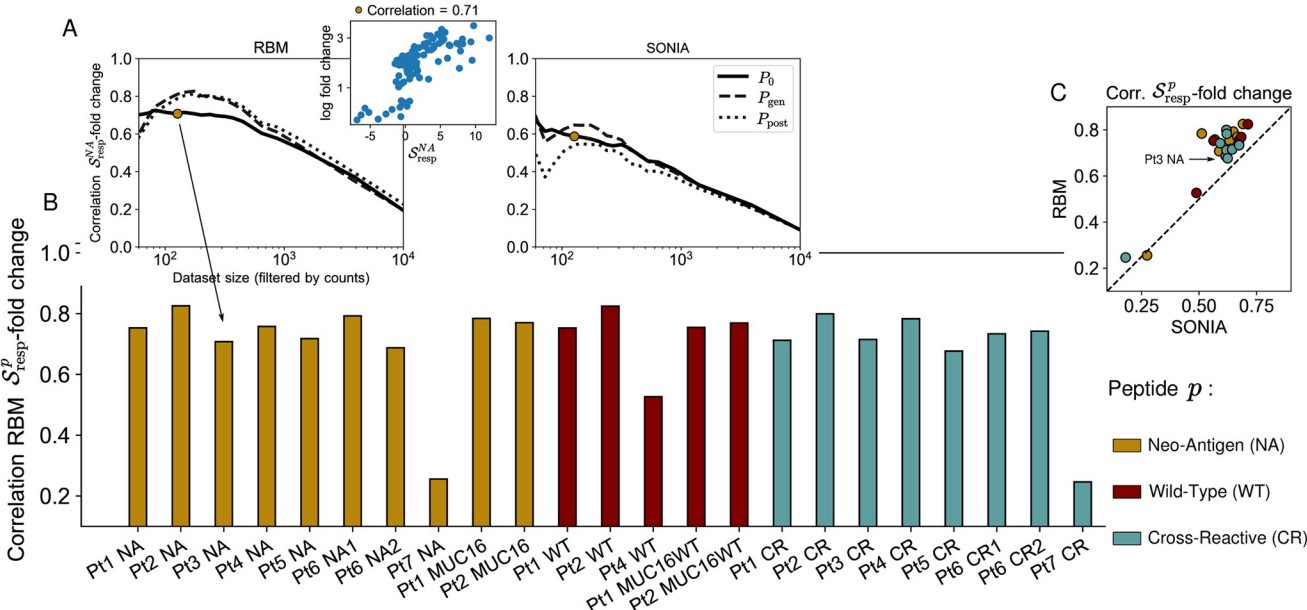

**Fig 3. Model scores of T-cell clonal expansion.** A: The RBM calculated response scores $\mathcal{S}_{\text{resp}}^{p}$ for each TRB clone in each antigen-stimulation condition $p$. The correlation between the $\mathcal{S}_{\text{resp}}^{p}$ and clone fold change is poor when considering all clones, as low-count clones are a source of noise. A significant correlation is recovered by progressively filtering out the low-abundance sequences from the testing set, shown on the example of stimulation by $p$ = NA of Pt3 PBMC. For both RBM and SONIA (shown on the right), the correlation is robust to the choice of different background distributions: $P_0$ (the probability learned on the same patient's dataset at day 0), $P_{\text{gen}}$ (the probability of generating a given CDR3, described by OLGA [21]) and $P_{\text{post}}$ (the post-thymic selection CDR3 distribution, sampled here from a default human TRB model available in SONIA [15]). The small differences between the 3 curves are mainly due to sampling, see also Materials and methods. B: The correlation coefficient between the RBM $\mathcal{S}_{\text{resp}}^{p}$ and clone fold change for each antigen-stimulation experiment $p$ (color-coded by antigen) when retaining the 125 most abundant clones (the golden dot in panel A). Data from [16] correspond to $p$ = NA, CR, WT, MUC16, MUC16WT (Pt1, Pt2), $p$ = NA, CR (Pt3, Pt5, Pt7), $p$ = NA, CR, WT (Pt4), $P$ = NA1, CR1, NA2, CR2 (Pt6), see also Table 1. C: Scatter plot comparing $\mathcal{S}_{\text{resp}}^{p}$ from the RBM and SONIA.

can be estimated relative to other "background" distributions at our disposal, accounting for TCR statistics in the peripheral blood in normal, unstimulated conditions. In Fig 3A, we show that the model prediction for antigen specific expansion at day 21, $\mathcal{S}_{\text{resp}}^{p}(\boldsymbol{\sigma})$, is robust with respect to the choice of the background distribution, suggesting we are capturing a response to this specific antigenic challenge.

## Model scores serve as predictors of specificity of T-cell response

The datasets considered portray the response of the same, individual TRB repertoire to different conditions determined by cultures with different antigens. We next asked whether the model is able to detect some condition-related specificity in these responses.

The model's response scores $\mathcal{S}_{\text{resp}}^{p}(\boldsymbol{\sigma})$ quantify differential degrees of expansion of a TRB clone $\boldsymbol{\sigma}$ in response to stimulation by antigen $p$ (S5A Fig). By comparing response scores to different antigens tested for the same patient, we define a score of specificity of response of clone $\boldsymbol{\sigma}$ to a given antigen $p$, $\mathcal{S}_{\text{spec}}^{p}(\boldsymbol{\sigma})$, see Eq 10 in Materials and methods. The score assigns positive values to TRB clones that are specific responders to the antigen and negative values to the clones unspecific to it, as shown for Pt3 in Fig 4A. Here, the expansion in response to the two tested antigens (the neoantigen NA, the cross-reactive CR) is mostly specific to each of them. Specificity scores whose values lie around zero denote both clones that did not expand at all and clones that expanded under both stimulations (cross-reactive clones), see Fig 4B. Cross-

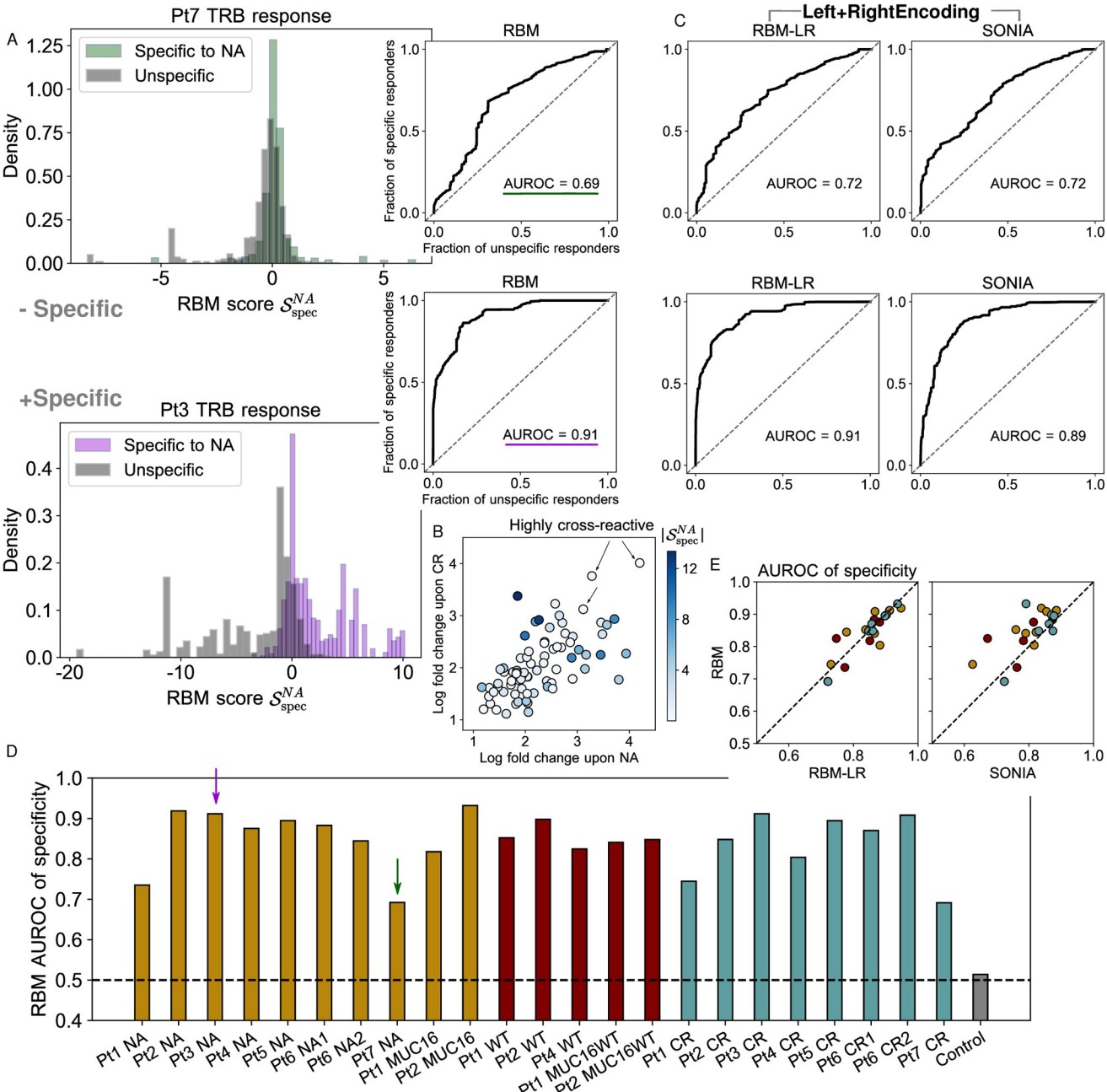

**Fig 4. Model scores of T-cell response specificity.** A: Histogram of RBM specificity scores $\mathcal{S}_{\mathrm{spec}}^{p}$ showing their ability to discriminate specific responders to $p$ = NA (assigned positive values) from unspecific ones (assigned negative values), for two examples: of less specific (Pt7 samples, top), and more specific expansion (Pt3 samples, bottom). ROC curves give the fraction of specific responders to $p$ = NA (clones specifically expanded upon NA stimulation) predicted by $\mathcal{S}_{\mathrm{spec}}^{NA}$ against the fraction of unspecific responders (clones responsive to the alternative stimulation, here CR) when varying the threshold of discrimination. The Area Under the ROC curve (AUROC) is taken as quantitative measure of specificity of expansion. B: Ref. [16] identified 78 clones that expanded under the stimulation by both NA and CR in Pt3. $x$ and $y$-axis give their log fold change in the NA and CR experiments. The color code gives their specificity score $\mathcal{S}_{\mathrm{spec}}^{NA}$ in absolute value (note that $|\mathcal{S}_{\mathrm{spec}}^{NA}| = |\mathcal{S}_{\mathrm{spec}}^{CR}|$). High $|\mathcal{S}_{\mathrm{spec}}^{NA}|$ is assigned only to clones whose expansion was more significant in one experiment than in the other, while clones that are significantly expanded in both experiments (highly cross-reactive) are assigned $|\mathcal{S}_{\mathrm{spec}}^{NA}|$ closer to zero. These clones do not contribute to discriminating the two repertoires. In A-B, $\mathcal{S}_{\mathrm{spec}}^{NA}$ and log fold change are given in log base 10. C: ROC for Pt7 and Pt3 samples obtained by RBM-LR and SONIA, which take input training data in the *Left+Right* encoding. D: AUROC calculated from the RBM for all datasets under consideration from [16]. The dashed line gives the expected value (AUROC = 0.5) when comparing statistically indistinguishable samples (the "control", gray bar), calculated from two replicates of Pt7 sample at day 0. E: Comparison of the AUROC obtained by the RBM, the RBM-LR model and SONIA for all datasets in D.

reactive clones are expected to arise due to the sequence similarity between CR and NA and were documented in Ref [16]. For Pt7, both specific and unspecific responders give similar distributions of specificity scores, due to a less marked difference in fold change under the two stimulations (Fig 4A), signaling that in Pt7 expansion is less specific.

We can set a threshold score value to discriminate specific from unspecific responders and, by varying this discrimination threshold, we can build a Receiver Operating Characteristic curve (ROC) describing the fraction of specific responders to the antigen predicted by the model's score, against the fraction of predicted unspecific responders. The AUROC (Area Under the ROC) is a quantitative indicator of both the specificity of the expansion upon exposure to a given antigen at the repertoire level and of the model's ability to detect such specific expansion (see Materials and methods). Fig 4D presents this measure for all samples, showing a high degree of specificity for most samples, and a lower degree specificity for others, in particular patient Pt7 (NA, CR) and Pt1 (NA, CR). This pattern suggests that the tested antigens, selected for each patient by *in silico* prediction, were able to stimulate specific responses in some patients more than in others, and the AUROC of specificity could help identify candidate immunostimulant neoepitopes.

When assessing specificity, we found that RBM and RBM-LR give consistent answers, see Fig 4C and 4E, as a consequence of the fact that the two methods' scores correlate in a similar way with clonal abundance. The excellent agreement between RBM and RBM-LR confirms the remarkable robustness of our scoring procedure to the choice of data encoding. AUROC of specificity by RBM/RBM-LR are also consistent with SONIA, albeit generally shifted towards higher values for RBM based approaches (Fig 4C and 4E), reflecting the increased ability of models that include correlations (RBM/RBM-LR) to predict clonal abundances (see also the inset in Fig 3 and S3 Fig).

## RBM-based dimensionality reduction of T-cell response

The RBM outputs a probability of response that is based on detecting sequence patterns associated to expansion (Fig 2A). The projections of sequences onto weights—the "inputs to hidden units" (see Eq 4 in Materials and methods) define a lower-dimensional representation space where data is structured based on sequence patterns (dimensionality reduction). Applied to the antigen-specific TRB repertoires from Ref. [16], this representation gives insight into how the response to the same neoantigen is distributed in sequence space, identifying clusters of similar clonotypes and separating out expansion "modes" due to dissimilar clonotypes. Focusing on the responding clones identified by RBM scores, visualization in 2 dimensions (Fig 5) highlights clusters of clones based on sequence features. The degree of clustering depends on the specific samples, *e.g.* it is more pronounced in the Pt5 CR sample than in the Pt3 NA sample. This dimensionality reduction exploits the interpretability of the RBM model in latent space.

To perform a comparative analysis of these sequence motifs, we applied GLIPH [22, 23], an algorithm that clusters TCR sequences based on putative epitope-specificity (S8 Fig). Most expanded clusters recovered by GLIPH can be associated with expanded sequences represented in RBM space in Fig 5. Beyond these common hits, GLIPH and RBM reveal complementary information, reflecting their different purposes. While GLIPH focuses on finding clusters of similar sequences, in our RBM approach we use information on expansion to appropriately weight sequences. As a result, GLIPH reports well-clustered but not necessarily expanded motifs, while RBM reports highly expanded but sometimes isolated sequences. Only sequences that are both well-clustered and expanded are captured by both algorithms.

To further illustrate the potential of the RBM low-dimensional representation, we considered human TCR validated to be specific to common viral epitopes by MHC tetramer-sorting

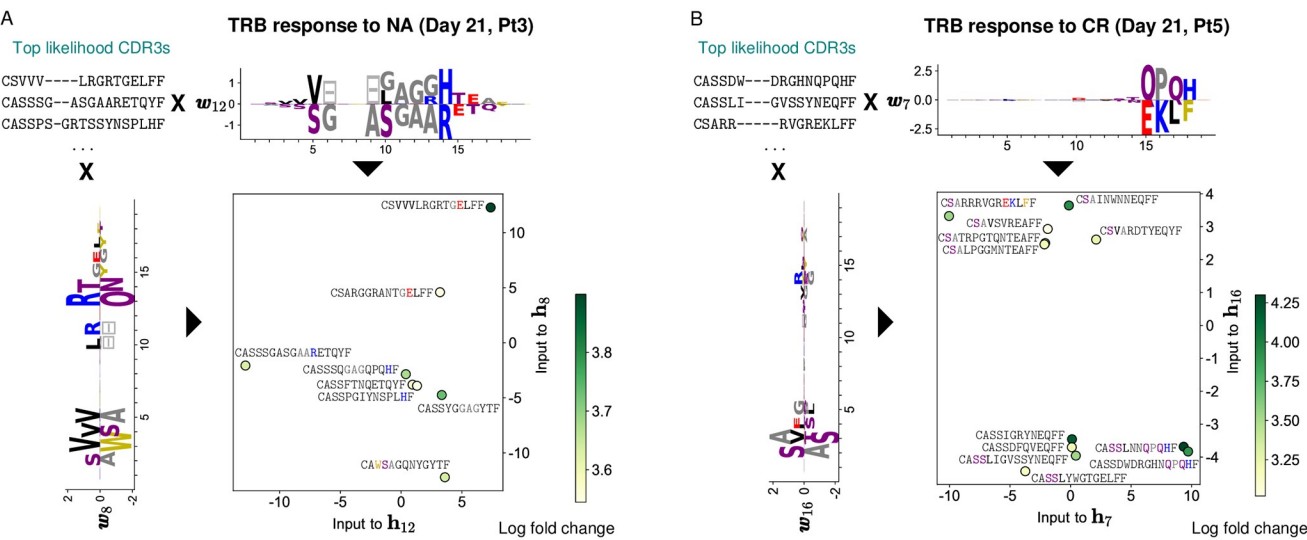

**Fig 5. RBM-based dimensionality reduction of TRB response.** A: A lower-dimensional representation of RBM predicted TRB sequences of Pt3 responding to the NA antigen 21 days post-stimulation (same model as in Figs 1 and 2). We rank CDR3 sequences by likelihood and look at the top ones (8 clones). We select 2 sets of RBM weights displaying well-defined patterns of amino acid usage at the beginning and at the end of the CDR3 (respectively $w_8$ and $w_{12}$). The projection of top-likelihood clones onto these weights define the inputs to the corresponding hidden units ($h_8$ and $h_{12}$) in two dimensions. Depending on sequence patterns (see colored amino acids), responding clones have either a positive or negative projection onto the selected weights and end up occupying different portions of the space of inputs to hidden units. The color code (log base 10 of fold change) highlights that high model likelihood reflects high fold change. B: Same data representation by the RBM as in A, with the 12 top-likelihood clones for the Pt5 CR assay.

[17, 22]. We tested the following epitopes: HLA-A*02:01-M1$_{58}$(M1$_{58}$) from the influenza virus, HLA-A*02:01-pp65$_{495}$ (pp65$_{495}$) from the human cytomegalovirus (CMV), and HLA-A*02:01-BMLF1$_{280}$ (BMLF1$_{280}$) from the Epstein-Barr virus (EBV) from both [17] and [22]; HLA-A*01:01-NP$_{44}$ (NP$_{44}$), HLA-B*07:02-NP$_{177}$ (NP$_{177}$) from Flu, HLA-B*07:02-pp65$_{417}$, and HLA-A*01:01-pp50$_{245}$ from CMV [22]. We trained two RBM on the full sets of CDR3 sequences from each of the two datasources [17] and [22] (see Materials and methods). The RBM-based dimensionality reduction identifies clusters of clones that specifically bind the M1$_{58}$, BMLF1$_{280}$, pp65$_{495}$ epitopes, for which characteristic sequence motifs were identified in Refs. [17, 22] using respectively the clustering algorithms TCRdist and GLIPH (Fig 6 and S7 Fig). RBM weights reproduce also three of the five sequence motifs identified by GLIPH and validated to be involved in response to *M. Tubercolosis* in [22] (S7(E) Fig).

## Repertoire diversity correlates with model generalizability

Individual repertoires are highly diverse and personalized. Even TRB subrepertoires specific to the same antigen show high diversity [6]. Several works have also highlighted that epitope-specific responses often have a "clustered" component, *i.e.* driven by groups of TCRs with common sequence motifs [17, 22, 24], along with a more "dispersed" component in sequence space [17]. To quantify the degree of heterogeneity of clone distributions within repertoires of interest, several standard diversity metrics exist [25, 26]. Here, we calculate an index of sequence dissimilarity that follows closely the logic of the repertoire diversity metric TCRdiv [17], but we apply it to sets of only CDR3 sequences (see Materials and methods) from the neoantigen experiments of Ref. [16]. The sequence dissimilarity index is a summary statistic that gives an effective measure of the heterogeneity among the responding (*i.e.* expanded) clones, which indicates how much the response is focused around certain amino acid patterns in sequence space. We see in general (Fig 7A) a higher CDR3 sequence dissimilarity for the

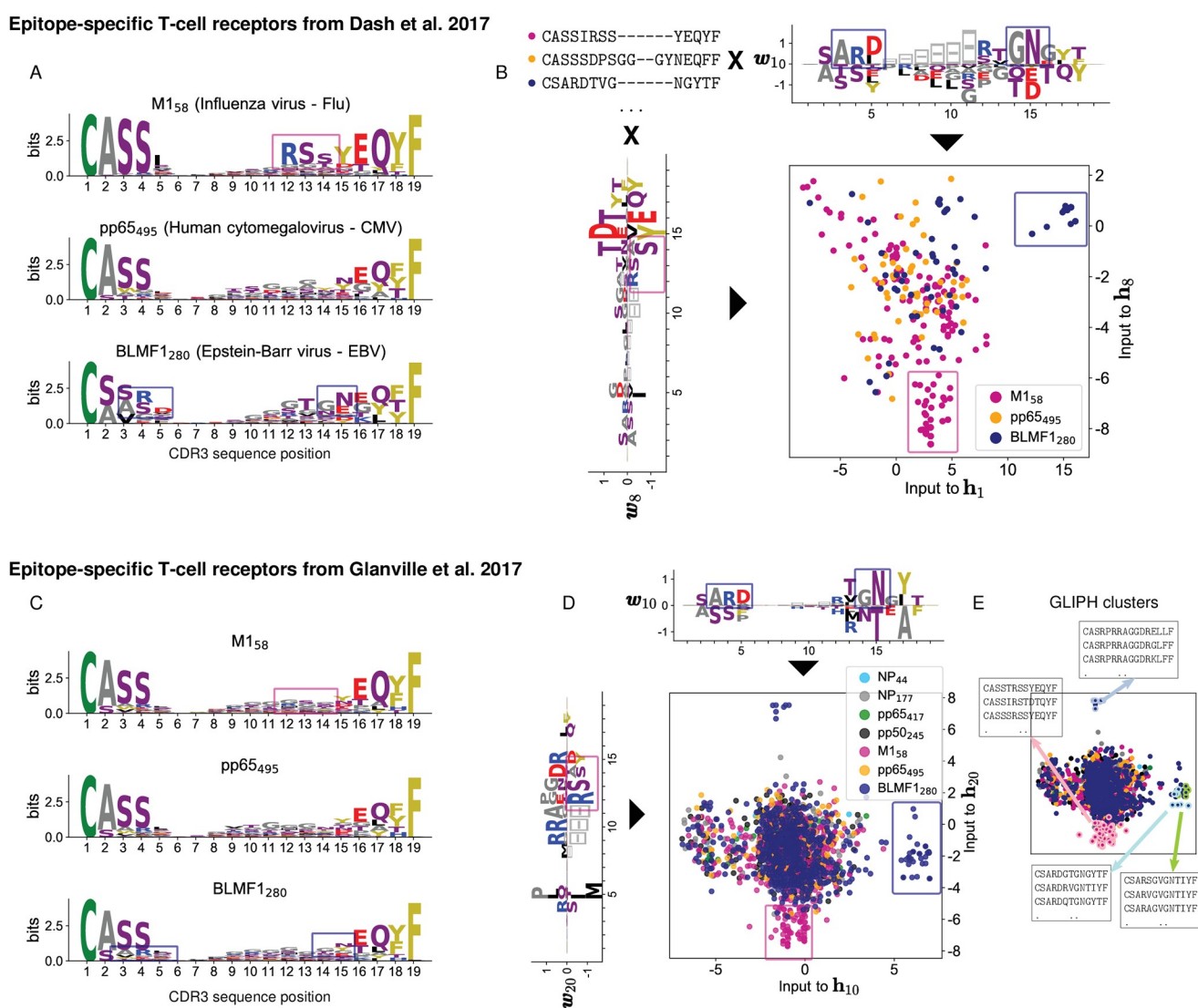

**Fig 6. RBM retrieves TRB sequence motifs reported in Dash et al. 2017 [17] and Glanville et al. 2017 [22].** A: Sequence logos of aligned CDR3 of each human TCR repertoire specific to CMV, EBV and Flu epitopes based on the tetramer sorted data from Ref. [17]. Sequence motifs identified in Ref. [17] are marked by colored boxes. The sequence motifs visualized here from *aligned* CDR3 are similar but not exactly the same as the ones in Ref. [17]. B: We learned an RBM on the full list of CDR3 sequences from the three repertoires in A. The learned weights show the same sequence motifs as in A and the inputs to the corresponding hidden units identify two groups of clones: one specific to the M1$_{58}$ epitope (magenta dots) and the other to the BMLF1$_{280}$ epitope (blue dots). C-D: Same representation as in A-B for the tetramer sorted data from Ref. [22] and the RBM trained on them. For the sake of comparison we have limited the logos in C to the same epitopes as in A. The pp65$_{495}$ repertoires (orange dots both in B and D) is the most heterogeneous (see Fig 7A) and the characteristic sequence motifs reported in [17, 22] are not easily visible in A-C (see instead S7A–S7C Fig). In both cases, the RBM is able to learn a set of weights reflecting such motifs, see S7A–S7D Fig. E: Same as D, but where T-cell clones belonging to 4 of the 35 clusters found by GLIPH in [22] are circled in color. These clones are well separated in RBM space. The RBM representation space captures the similarity of sequences that have the same epitope specificity but are placed into different clusters by GLIPH (blue- and green-circled clusters).

neoantigen response than for epitope-specific repertoires from Refs. [17, 22] (in particular the M1$_{58}$ repertoire from [17]), indicative of a more diverse response.

Shared sequence patterns play a crucial role in enabling prediction of antigen-specific response from sequences alone [17, 22, 27]. We can hypothesize that high sequence dissimilarity can hinder generalization, since it is based on picking up features of responding sequences that are not present in the non-responding ones. In fact, the sequence dissimilarity index for

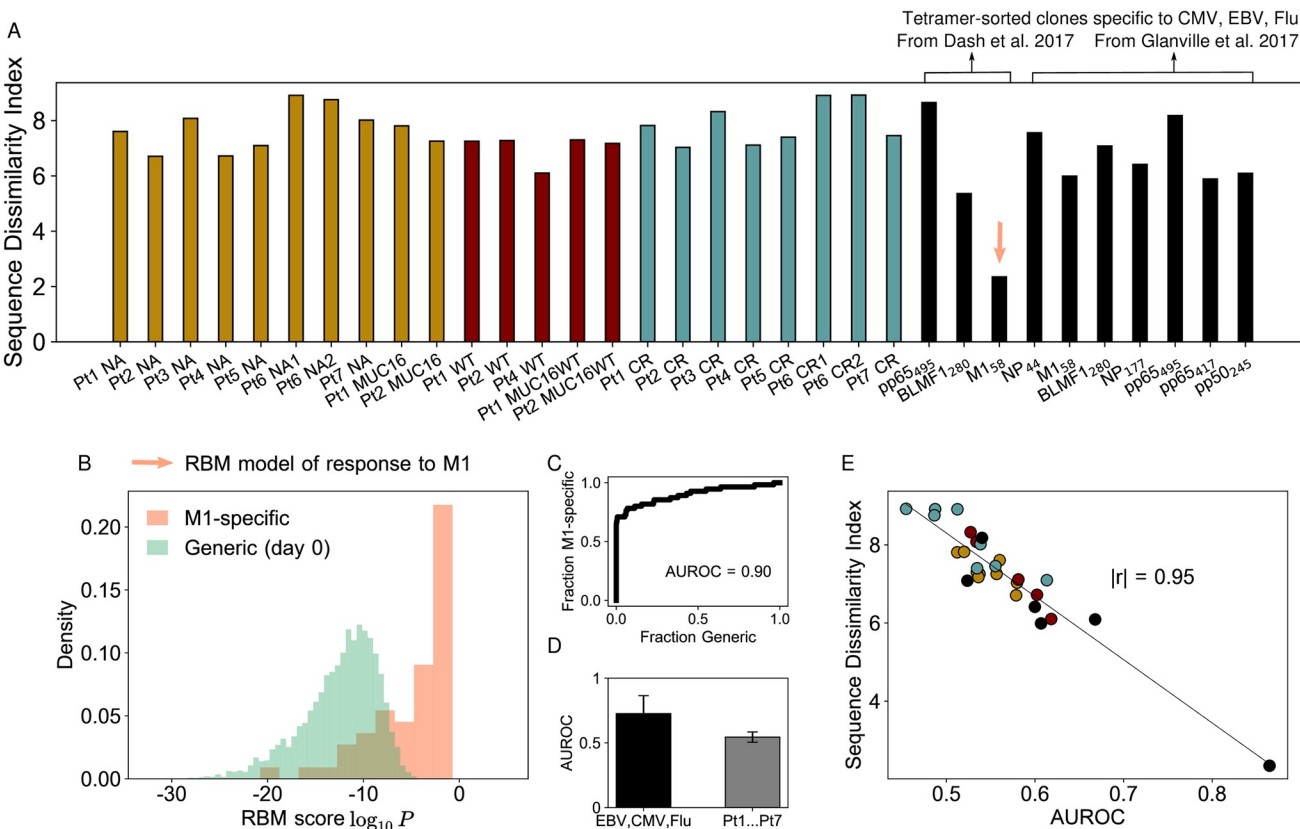

**Fig 7. Diversity among responding clones and model's predictive power.** A: The sequence dissimilarity index (a modified TCRdiv diversity index, see Materials and methods) calculated for the pool of expanded clones in the Pt1,. . .,Pt7 assays from [16]. The dissimilarity index values lie in a range comparable only to the most diverse epitope-specific human repertoire of tetramer-sorted TCRs from Dash et al. [17] and Glanville et al. [22] (black bars), see also Fig 6. B: Probabilistic scores from the RBM (trained on the M1-specific repertoire from [17]) evaluated for a testing set of M1-specific CDR3 and generic CDR3 randomly drawn from the day 0 of Pt1,. . .,Pt7 samples. The model's ability to predict M1-specific clones is quantified by the high AUROC value (C). D: AUROCs for models trained on lower-diversity repertoires (tetramer-sorted TCR data from [17, 22], black bar) are in average higher than for the Pt1,. . .,Pt7 datasets (gray bar). Bar plots give the average AUROC and its standard deviation. AUROC values are estimated by a leave-one-out 5-fold validation protocol (see Materials and methods). E: AUROC for the tetramer-sorted and Pt1,. . .,Pt7 datasets is inversely correlated with the sequence dissimilarity index, Pearson correlation $r$ of magnitude $|r| = 0.95$, $p$-value for testing non-correlation $= 2.53 \times 10^{-14}$ ($|r| = 0.87$ and $p$-value $= 1.9 \times 10^{-7}$ considering only Pt1,. . .,Pt7 datasets). D-E show that the predictive power of sequence-based models is degraded for high-diversity repertoires. We discarded from the AUROC test of B-E the pools of responding clones consisting of fewer than 100 sequences (Pt6 NA1, BMLF1$_{280}$ and pp65$_{495}$ from [17], NP$_{44}$ and pp65$_{417}$ from [22]).

the neoantigen data (Fig 7A) is comparable in magnitude only to the most diverse epitope-specific human TRB repertoire from [17], for which the distance-based clustering algorithm performed worse than for other repertoires. To test more quantitatively this hypothesis, we built an RBM model for the M1-specific TRB repertoire from Ref. [17], the one with the lowest sequence dissimilarity index (Fig 7A). We evaluated the model on a held-out test set of M1-specific clones and a 99-fold excess of "generic", unspecific ones (CDR3 sequences randomly drawn from the day 0 samples). By comparing the RBM scores assigned to M1-specific clones and to generic ones (Fig 7B), we found the model performed extremely well at predicting clones reactive to M1 epitopes in terms of the Receiver Operating Characteristic curve and its associated area (Fig 7C), with AUROC = 0.9 on 5-fold leave-one-out validation (see Materials and methods). Repeating the same test using sets of responding clones with a much higher sequence dissimilarity index to train our RBM, such as the neoantigen stimulation datasets from Ref. [16] (restricting to expanded clones) or the repertoires specific to pp65$_{495}$, BMLF1$_{280}$

from [22], we found that the average model performance is significantly decreased (Fig 7D). Since tetramer-sorted datasets do not include count information, this performance reflects the ability to predict unseen sequences, which is a more stringent test than predicting the abundance of observed sequences. A similar test can be performed on the PBMC stimulation experiments from [16] by separating TCR sequences into training and testing sets, in such a way that a given sequence could not be found in both. Doing so reveals relatively poor predictability (S9 Fig), which is due to the very high diversity of responding clonotypes in these datasets. In general, we verified that the model's ability to recover specific clones is inversely correlated with CDR3 sequence dissimilarity (Fig 7E and S9(A) Fig).

## Discussion

We have proposed probabilistic-modeling approaches (summarized in Fig 8A) to obtain sequence representations of T-cell response from RepSeq datasets. We demonstrated the inferred, probabilistic scores capture information about sequence counts and predict clonotype fold changes upon antigen-stimulation (Fig 3). These methods provide tools to detect specific TCR responses (Fig 4) and connect information about clonal expansion to sequence features in a way that enables the inspection of sequence properties necessary for TCR recognition at the biochemical level, providing important input for experimental validation (Figs 2 and 5).

Overall we found the RBM model is better at capturing information about the response to a single neoantigen or epitope stimulation contained in clone counts than SONIA (Fig 8B–8C). The RBM encodes global correlations in the CDR3 while SONIA is an independent-site model (Figs 3C and 4E, and S3 Fig). Nevertheless, SONIA is able to incorporate some sequence correlations via $P_{gen}$ and account for the V-J gene segments flanking the CDR3, which already improve predictions in comparison to a fully factorized probabilistic model (S6 Fig). The data set we have analyzed [16] is characterized by very dissimilar clonotypes. Such diversity results in relatively poor RBM performance in predicting responding clonotypes not included in the training data sets in the leave-one-out validation protocol (S9 Fig). SONIA performance is correlated to RBM performance (S9(C) Fig), albeit still poor, supporting that the ability to generalize to unseen clonotypes is weakly dependent on the model and data encoding chosen. Rather it is largely determined by the heterogeneity of the repertoire. In general, the predictive power of sequence-based models is largely improved when they are trained on less diverse data sets as tetramer data from the M1-specific repertoire from Ref. [17] (see Fig 7D).

The RBM also has the ability to project the response onto different (groups of) clonotypes (Fig 5), which is useful for identifying the different sequence features underlying response within the same sample. Combining different datasets describing the response to specific antigens gives hope of linking the specificity of response to distinct TRB sequence features. Any generative approach for the sequence probability could be used, for example VAE models [28, 29]. While for fitting post-thymic selection TCR distribution methods capturing non-linearities did not seem to be essential (there VAE performs as well as SONIA [30]), for building models of TCR response we expect the VAE to reach an accuracy similar to RBM.

Identifying TCR reactivity to an antigen that is a single mutation away from a self-antigen (neoantigen), but that is not reactive to the self-antigen itself (the corresponding wild-type), is of major importance for personalized cancer immunotherapies, such as vaccine design. Candidate neoantigens to prioritize in vaccine design are selected through time- and resource-consuming pipelines that involve several steps of neoantigen identification and experimental validation. These approaches are often too costly and time-consuming to carry out on a single-patient basis. Both immunodominant neoantigens and the elicited immune responses are in

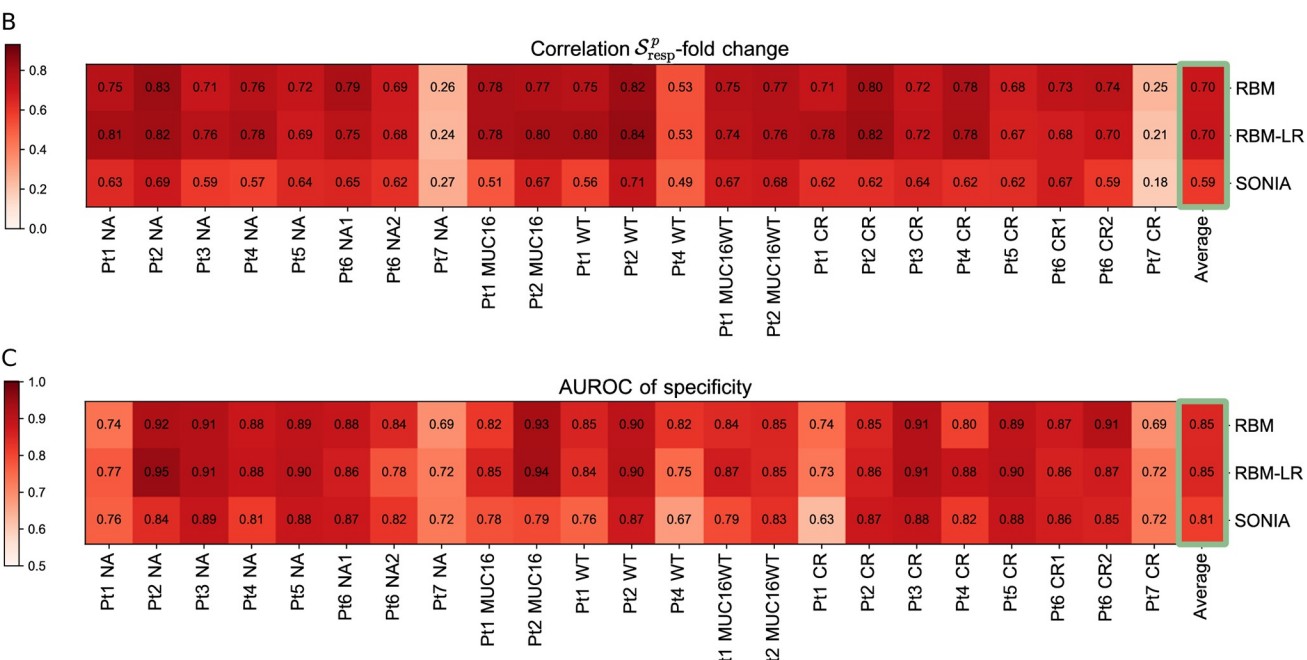

**Fig 8. Comparison of the methods.** A: Schematic summary of the characteristics of the three sequence-based probabilistic modeling approaches used: RBM, RBM-LR, SONIA. B-C: Summary of the results we obtained for the samples from [16] using the approaches in A, in particular the correlation between the model's response score $\mathcal{S}^p_{resp}$ and TRB clone fold change when retaining the 125 most abundant clones (B) and the AUROC-based measure of response specificity (C). Both these measures highlight a different degree of specific response to the stimulation, that overall stays high across the samples apart from Pt7, where response to the peptides tested was rather unspecific. For each sample, the results from the three methods are well in agreement (see also Figs 3C and 4E). High values of the AUROC and of the $\mathcal{S}^p_{resp}$-fold change correlation reflect also how well the probability inferred by each method reproduces the clone abundance (see S3 Fig, Materials and methods), thus we report the averages over all samples (last column) for a comparison of the methods' performance. This comparison shows that the general performance is higher for RBM-based approaches than for SONIA and is the same for RBM and RBM-LR.

fact largely individualized, hence the information from already analyzed patients is in general insufficient to probe candidate neoantigens for new patients (unless we consider recurrent cancer mutations, which could provide shared targets across patients and tumor types but whose availability remains to be assessed [31]). In this setting, approaches such as the RBM and SONIA can help validate specific responses to pre-selected neoantigens, where they can be applied to extract the sequence features of neoantigen-specific responses in individual and newly-produced datasets. Moreover, the inferred probabilistic score renders our models generative, in principle, implying one can predict new TCRs that recognize an antigen, a point critical to the ability to engineer T cells with a given specificity. It would be interesting to validate, by assays probing the response to neoantigens and wild-type antigens, to what extent our modeling approach is able to distinguish the wildtype-specific receptors from the neoantigen-specific receptors, and whether one can then generate new TCRs with comparable specificity.

Our approach is designed to have broader applicability. For instance we have benchmarked the model with datasets describing individual repertoires elicited by different known antigens,

without using any information about the response-epitope association. This shows that our approach is generally applicable to scenarios where only single or multiple snapshots of TCR repertoires (and not the stimulating epitopes) are available, either from peripheral blood or from tissue-infiltrating lymphocytes. In these cases, our method can assess whether TCR-driven immune response is occurring and how specific it is across different conditions, time points, and tissues. It could be used to discriminate clones specifically responding to different, unknown targets or to detect changes in repertoire clone composition at different stages of disease progression. Here we exploited the longitudinal structure of the available datasets to assess TCR response after stimulation relative to the pre-stimulation time point, but such longitudinal structure is not strictly necessary, as we have shown in Fig 3A. Our approach can be applied to single time point datasets, where it can characterize TCR response and its specificity in comparison to reference distributions for normal, unstimulated TCR repertoires, like the ones provided by existing probabilistic models [15, 21]. In addition, in Figs 6 and 7B–7E we show an application of the RBM based model to datasets of unique, only responding clones identified by direct TCR binding to MHC-tetramer reagents where a non-responding dataset is not available.

Our approach remains designed to analyze the ongoing response to known or unknown targets in TCR repertoires. It does not have the power to predict the number and the type of such targets and how in turn they affect the amount of well-clustered and expanded clones. In tetramer-sorted, viral epitope-reactive TCR data from [17, 22], we observed that the RBM low-dimensional representation groups together into well visible clusters sequences sharing defined amino acid motifs. In the majority of cases, different groups corresponded to responses to different epitopes. In general, however, determining the correspondence between clustered CDR3 sequences and targets of response is outside of the scope of our method, since it would require better knowledge of the actual TCR-peptide binding modes and, in cancer, of the tumor's characteristics that are susceptible to produce viable neoantigens. With more data on antigen-specific responses becoming available, it will be interesting to understand how biophysical properties of antigens correlate with the degree of clustering in the triggered TCR response.

Another related limitation is the fact that the RBM does not disentangle clusters of clones with common amino acid patterns from isolated clones that are expanded but cluster less well. A procedure that separates out well-defined clusters, and provides typical predictions of a clustering algorithm, would overcome this limitation and could be directly compared to GLIPH [22] and TCRdist [17]. The statistical significance of the feature enrichment captured by clusters could be further enhanced by directly learning an RBM with a "differential" structure, akin to SONIA's structure, where the weights, similarly to $q$-factors, are allocated to learn the differences with respect to a baseline repertoire distribution due to a targeted immune response.

Errors in the inference procedure can arise due to incorrect estimates of clone abundances in data without unique molecular barcodes [6, 32]. Low-count sequences, lying a few mutations away from responding clones, may represent variants of high-frequency clonotypes arising from sequencing errors and could be discarded by setting thresholds in counts below which clonotypes are filtered out. We used a count-based filtering threshold only when correlating model scores to clonal abundances (Fig 3 and S3 Fig). We used a variable threshold parameter, avoiding having to estimate a precise cutoff for clone expansion from counts. Such estimation can depend on count sensitivity to experimental conditions and sequencing protocols and it requires to first infer the expected level of noise from replicates of the experiment [11]. Replicates were not available here, as is often the case for RepSeq data. When replicates are available, the choice of a more systematic count-based filtering procedure can be used

using a probability of expanded clonotypes [11]. Re-weighting sequences by their probability of expansion could provide also a robust solution to correct for these biases and constitutes an avenue for future work.

In the neoantigen stimulated data, we observed, perhaps unsurprisingly, a lower degree of repertoire focusing around a few, similar TRB clones than in examples of epitope-specific repertoires from MHC-tetramer assays for viral antigens (Fig 7). We cannot exclude that this effect is due to the type of assay where, differently from MHC-tetramer assays, responding clones are not isolated by direct binding to the antigen. Only some of the responding clones may be antigen-specific, while others may have expanded in response to other, not directly controlled, culture conditions. Additional experimental tests are necessary to disentangle the heterogeneity of response induced by different molecular targets and the heterogeneity of response due to other recognition modes of the same epitope. If the response described here is not antigen specific, it still remains the response detected by clonal expansion and our models describe a *condition*-specific rather than *epitope*-specific response. This difference in interpretation is important biologically, and from the view of setting up future experimental assays, but it does not impact the presented methodological results.

## Materials and methods

### Dataset preparation

**Data pre-processing.** The main dataset studied consist of TRB CDR3 regions sequenced by the Adaptive Biotechnologies ImmunoSEQ platform from the T-cell assays of Ref. [16]. We discard sequences that are generated by non-productive events, *i.e.*, they do not have the conserved anchor residues delimiting the CDR3 region (Cysteine as the left anchor residue, Phenylalanine or Valine as right one) and they align to pseudo-genes as germline gene choices. We collapse sequences with the same CDR3 composition (including sequences with different V-J genes) into the same clonotype and assign, to each clonotype, a multiplicity (number of sequence reads) that sums up multiplicities of all the different DNA sequences coding for that same CDR3. The fraction of reads with the same CDR3 but different V-J genes was small (in average 0.06).

**Characterization of TRB clone expansion.** In the PBMC culture stimulated by antigen $p$, a CDR3 sequence $\boldsymbol{\sigma}$ is detected with a multiplicity that we denote by $N_t^p(\boldsymbol{\sigma})$, at time $t = 21$ days. We define the fold change $FC^p(\boldsymbol{\sigma})$ for a sequence $\boldsymbol{\sigma}$ in the repertoire measuring response to antigen $p$ as:

$$FC^p(\boldsymbol{\sigma}) = \frac{N_{21}^p(\boldsymbol{\sigma}) + 1/2}{N_0(\boldsymbol{\sigma}) + 1/2},$$

(1)

where $N_0(\boldsymbol{\sigma})$ is the multiplicity of sequence $\boldsymbol{\sigma}$ in the blood sample from the corresponding patient at day 0 (before any antigen stimulation). In Eq 1, we add to all counts a 1/2 pseudo-count, as a standard procedure to avoid ill-defined fold change with very low counts [33].

To isolate a pool of expanded clones for each stimulation condition, we follow the same criteria as Balachandran et al. [16], *i.e.* TRB clones that increased > 2-fold on day 21 compared to day 0 and fulfilled the Fisher's exact test and Storey's Q value for false discovery rate were defined as expanded.

**CDR3 sequence alignment.** We first trained an RBM with fixed-length CDR3 sequences. To this end, we built a multiple sequence alignment (MSA) of $N^\sigma$ sites of all the 276993 unique CDR3 amino acid sequences in the combined dataset from all assays. The alignment routine consists of aligning in a progressive way sequence profiles of the same length, as schematically summarized by the following steps (see also Ref. [19]):

1. We build reference profiles for each length $l$ by estimating the Position Weight Matrix ($PWM_l$) for the subset of CDR3s with length $l$ by the routine *seqprofile* (with default options) of the Matlab Bioinformatics Toolbox (release R2018b).

2. We align progressively fixed-length sequence profiles $PWM_l$ starting from the minimal length $l = l_{min}$ (here $l_{min} = 5$ residues) to an upper length value $l = l_{max}$ (here we set $l_{max} = 23$ residues, length below which 99.9% of sequences lie, see S1 Fig) by aligning at each step two profiles differing by one residues in length. We use the function *profalign* of the Matlab Bioinformatics Toolbox, with BLOSUM62 as scoring matrix. This procedure results in progressive gap insertions in the shorter CDR3s and produces an alignment of length $l_{max}$.

3. We use the alignment obtained as seed to learn a Hidden Markov Model (HMM) profile of length $N^\sigma$, using the routines *hmmprofstruct* and *hmmprofestimate* (with default parameters) of the Matlab Bioinformatics Toolbox. We have chosen $N^\sigma = 19$ since most sequences ($\sim 98\%$) have length below this value, see S1 Fig.

4. We align CDR3 sequences of length $l \neq N^\sigma$ to the HMM profile exploiting the position-specific HMM insertion and deletion probabilities (via the *hmmprofalign* function) to obtain a final MSA of length $N^\sigma$.

## Sequence-based probabilistic models

**RBM.** An RBM [12, 13] is a graphical model composed of one layer of $N^\sigma$ observed units $\boldsymbol{\sigma} = \{\sigma_i\}_{i=1}^{N^\sigma}$ and one layer of $N^h$ hidden units $\boldsymbol{h} = \{h_\mu\}_{\mu=1}^{N^h}$ connected by weights $\boldsymbol{W} = \{w_{i\mu}\}$ (Fig 1D). The observed units $\boldsymbol{\sigma}$ represent CDR3 sequences, hence the number of observed units for each sequence input is $N^\sigma = 19$ and each observed unit can assume $q = 21$ values (the 20 amino acids and the gap). The full probability distribution defining the RBM is parametrized as a joint probability over hidden and observed units:

$$P(\boldsymbol{\sigma}, \boldsymbol{h}) \sim \exp\left(\sum_{i=1}^{N^\sigma} g_i(\sigma_i) - \sum_{\mu=1}^{N^h} \mathcal{U}_\mu(h_\mu) + \sum_{i,\mu} h_\mu w_{i,\mu}(\sigma_i)\right), \tag{2}$$

where $g_i(\sigma_i)$ are $N^\sigma \times q$ local potentials acting on observed units, $\mathcal{U}_\mu(h_\mu)$ are $N^h$ local potentials on hidden units and the weights $w_{i,\mu}(\sigma_i)$ are $N^\sigma \times N^h \times q$ parameters coupling hidden and observed units with a strength dependent of the amino acid $\sigma_i$, see Figs 2A and 5. The probability distribution of CDR3 sequences can be retrieved as the marginal probability over hidden units:

$$P(\boldsymbol{\sigma}) = \int \prod_{\mu=1}^{N^h} dh_\mu P(\boldsymbol{\sigma}, \boldsymbol{h}) \sim \exp\left(\sum_{i=1}^{N^\sigma} g_i(\sigma_i) + \sum_{\mu=1}^{N^h} \Gamma_\mu(I_\mu(\boldsymbol{\sigma}))\right), \tag{3}$$

where $\Gamma_\mu(I(\boldsymbol{\sigma})) = \log \int dh\, e^{-\mathcal{U}_\mu(h)+hI}$. The input to the hidden unit $\mu$, $I_\mu(\boldsymbol{\sigma})$, coming from the observed sequence $\boldsymbol{\sigma}$ is given by:

$$I_\mu(\boldsymbol{\sigma}) = \sum_i w_{i,\mu}(\sigma_i). \tag{4}$$

We take $\mathcal{U}_\mu(h_\mu)$ in the form of a double Rectified Linear Unit (dReLu) potential:

$$\mathcal{U}_\mu(h) = \frac{1}{2}\gamma_{\mu,+}h_+^2 + \frac{1}{2}\gamma_{\mu,-}h_-^2 + \theta_{\mu,+}h_+ + \theta_{\mu,-}h_- \quad h_+ = \max(h,0) \quad h_- = \min(h,0), \tag{5}$$

a form that gives the model high expressive power, allowing to fit higher order correlations in the input data, see [14, 34]. All RBM parameters, weights $w_{i,\mu}(\sigma_i)$, local potentials $g_i(\sigma_i)$ and the parameters specifying $\mathcal{U}_\mu(h_\mu)$—here ($\gamma_{\mu,+}$, $\gamma_\mu$, $\theta_{\mu,+}$, $\theta_{\mu,-}$)—are inferred from data by maximization of the average likelihood $\mathcal{L} = \langle \log P(\boldsymbol{\sigma}) \rangle_{\text{data}}$ of sequence data $\boldsymbol{\sigma}$ through a variant of stochastic gradient ascent, as described in [19]. In this way, the RBM probability distribution is inferred in such a way as to optimally explain the statistics of our count-weighted training datasets—see S4E and S4H Fig for comparison of data and model single-site frequency and pairwise connected correlations. Penalty terms (regularizations) can be introduced in the log-likelihood maximized during training to prevent overfitting. Following [14], we use a $L_1$-type regularization:

$$\mathcal{L} - \frac{\lambda_1^2}{2q\,N^\sigma} \sum_\mu (\sum_{i,\nu} |w_{i,\mu}(\sigma)|)^2, \tag{6}$$

where $\lambda_1^2$ is the regularization strength.

For RBM models learned from count-weighted datasets (Figs 3–5), we use the performance of the RBM likelihood $\mathcal{L} = \log P_{21}^p(\boldsymbol{\sigma})$ at recovering the fold-change increase on the validation set (a held-out 20% of data) to search for optimal values of regularization ($\lambda_1^2$) and the number of hidden units ($N^h$), see S4C and S4D Fig. For the RBM models inferred from sets of only responding clones (Figs 6 and 7) the search for optimal parameters has been carried out in a standard way, *i.e.* by monitoring overfitting in the validation set at different values of $\lambda_1^2$ and $N^h$, see S4F and S4G Fig. Based on these searches, all the RBM models trained on count-weighted datasets in this work have been trained with 25 hidden units and regularization strength $\lambda_1^2 = 0.1$, the RBM models trained on datasets of only responding clones have been trained with $N^h = 10$ and $\lambda_1^2 = 0.1$ (exceptions are the RBMs learnt on TRBs from [22] either specific to CMV, EBV, Flu or to *M. Tubercolosis*, where we used $N^h = 25$ to better recover the different specificity-related motifs). The software package used for implementing the RBM model is described in [19].

**SONIA.** SONIA is a tool of probabilistic inference from TRB sequence data, where the inferred probability of seeing a given TRB sequence in the studied repertoire is parametrized in terms of an independent-site model:

$$P(\boldsymbol{\sigma}) = Q(\boldsymbol{\sigma})P_{\text{gen}}(\boldsymbol{\sigma}) \quad Q(\boldsymbol{\sigma}) = \frac{1}{Z}\prod_{i=1}^l q_{i,l}(\sigma_i), \tag{7}$$

where we take $\boldsymbol{\sigma}$ to represent only the CDR3 amino acid sequence, of length $l$, of the TRB chain. $P_{\text{gen}}(\boldsymbol{\sigma})$ is the probability of observing sequence $\boldsymbol{\sigma}$ in the repertoire as a result of the TCR generation process [21, 35]. $P_{\text{gen}}(\boldsymbol{\sigma})$ is computed by SONIA with the aid of a dedicated software package, OLGA [21]. $q_{i,l}(\sigma_i)$, with the index $i$ running over the CDR3 sites up to its length $l$, is the set of position-specific $q$-factors and $Z$ is the normalization constant. Using an encoding into *Left+Right* features, we consider one representative sequence length $l = N^\sigma = 19$ and each $q$-factor is estimated, for a given amino acid $A$, as the product:

$$q_{i,N^\sigma}(A) = q_i^L(A) \times q_{i-N^\sigma-1}^R(A), \tag{8}$$

where the left (superindex $L$) and right (superindex $R$) terms carry information about the position $i$ of the amino acid $A$ from the left and on the position $i - N^\sigma - 1$ from the right (see Fig 2B for an example). $q$-factors are learned by maximum likelihood estimation (achieved through rejection sampling [15, 36]) aimed at best reproducing by $P(\boldsymbol{\sigma})$ the sample statistics, see S4(B) Fig. An $L_2$ regularization, controlling the magnitude of the $q$-factors learned, can be

included in the maximum likelihood estimation and we set its value based on a hyperparametric search, see S4(A) Fig. SONIA models were learned and evaluated only for CDR3 sequences at $P_{\text{gen}} \neq 0$, where we estimated $P_{\text{gen}}$ by OLGA on the CDR3 only (without conditioning on the variable and joining gene segment) for consistency with our definition of a TRB clone.

**RBM-LR.** In the RBM-LR version, as a result of the *Left+Right* encoding, the RBM-inferred parameters consist of the superposition of a left and a right contribution, similarly to $q$-factors in Eq 8. For instance, the local potential $g_i(A)$ acting on amino acid $A$ at position $i$ is given by $g_i^L(A) + g_{i-N^\sigma-1}^R(A)$. For a given hidden unit $\mu$, the effective weight contributing to the activation of $\mu$ for amino acid $A$ at position $i$ is given by the sum $w_{i,\mu}^L(A) + w_{i-N^\sigma-1,\mu}^R(A)$, see S2 Fig. For the sake of comparison, we choose for the RBM-LR the same architecture as for the RBM (25 hidden units, regularization strength $\lambda_1^2 = 0.1$). Due to the *Left+Right* encoding, the length of the effective sequences used for training reaches several hundreds of sites: choosing the reference sequence length of $N^\sigma = 19$ (but shorter reference lengths could be used, see [15]), the overall length is $19 \times 20 \times 2 = 760$ (20 is the number of amino acids while 2 accounts for having parameters for both left and right alignment). This number, compared to the pre-aligned, fixed-length training datasets of RBM, can increase significantly the training time: a training of 200 iterations using 1 core lasts $\sim$ 2 hours with a CDR3 sample arranged into a multiple sequence alignment of $N^\sigma = 19$, and $\sim$ 11.7 hours for the same sample in the *Left+Right* encoding with $N^\sigma = 19$. Note that the RBM package includes an option for parallelization over multiple cores, which would help reduce the elapsed real training time.

## Scoring T-cell response

We train probabilistic models (RBM, SONIA) on count-weighted datasets at $t = 0, 21$ for each antigen $p$. For a given probabilistic framework (RBM, SONIA), we denote the probability assigned to sequence $\boldsymbol{\sigma}$ by the model trained on the repertoire snapshot at time $t$ under stimulation by epitope $p$ as $P_t^p(\boldsymbol{\sigma})$. We define the response score of a given CDR3 clonotype sequence $\boldsymbol{\sigma}$:

$$\mathcal{S}_{\text{resp}}^p(\boldsymbol{\sigma}) = \log\left(\frac{P_{21}^p(\boldsymbol{\sigma})}{P_0(\boldsymbol{\sigma})}\right) \tag{9}$$

the model likelihood assigned to sequence $\boldsymbol{\sigma}$ at day 21 relative to the one at day 0 in the same patient. The term $P_0(\boldsymbol{\sigma})$ can be in principle replaced by the probability provided by other "background" models. For instance one can take such background statistics as the distribution $P_{\text{gen}}(\boldsymbol{\sigma})$ [21] or the distribution $P_{\text{post}}(\boldsymbol{\sigma})$ from the models of thymic selection learned by SONIA [15] (and available with its package), see Fig 3A. When using $P_{\text{gen}}$ and $P_{\text{post}}$ to estimate $\mathcal{S}_{\text{resp}}^p$ (see Fig 3A), we assume that no read count information before stimulation is available, hence we set $N_0 \sim 1$, implying $FC^p \sim N_{21}^p$. As such, $P_{\text{gen}}$ and $P_{\text{post}}$ are effectively less noisy version of $P_0$, because sequences that sample $P_{\text{gen}}$ and $P_{\text{post}}$ can be generated in large numbers ($10^6$) and they are not affected by experimental sampling noise (as instead samples at day 0 are).

## Scoring specificity of T-cell response

To test the power of a model trained on the NA sample to discriminate responses that are specific to it from the ones that are specific to the CR sample, we calculate a specificity score of response to NA as:

$$\mathcal{S}_{\text{spec}}^{NA}(\boldsymbol{\sigma}) = \mathcal{S}_{\text{resp}}^{NA}(\boldsymbol{\sigma}) - \mathcal{S}_{\text{resp}}^{CR}(\boldsymbol{\sigma}). \tag{10}$$

If the responses to NA and to CR are specific, we expect $\mathcal{S}_{\text{spec}}^{NA}(\boldsymbol{\sigma}) > 0$ for $\boldsymbol{\sigma}$ in the set of sequences such that $FC^{NA}(\boldsymbol{\sigma}) > FC^{CR}(\boldsymbol{\sigma})$ and $\mathcal{S}_{\text{spec}}^{NA}(\boldsymbol{\sigma}) < 0$ for $\boldsymbol{\sigma}$ such that $FC^{CR}(\boldsymbol{\sigma}) > FC^{NA}(\boldsymbol{\sigma})$, as is seen in the histograms of Fig 4A for Pt3. When, for the same individual, we have samples for several antigens $p$ (Pt1, Pt2, Pt4, Pt6, see Table 1), we take as specificity score of response to a given epitope $p$:

$$\mathcal{S}_{\text{spec}}^{p}(\boldsymbol{\sigma}) = \mathcal{S}_{\text{resp}}^{p}(\boldsymbol{\sigma}) - \max_{p'} \mathcal{S}_{\text{resp}}^{p'}(\boldsymbol{\sigma}), \tag{11}$$

where the max is taken over the antigens $p' \neq p$ tested for the same individual. In this case, a sequence $\boldsymbol{\sigma}$ is considered a specific responder to epitope $p$ if $\max_{p'} FC^{p'}(\boldsymbol{\sigma}) = FC^{p}(\boldsymbol{\sigma})$, while it is considered as an unspecific responder to the antigens $p' \neq p$ (see the distribution of $\mathcal{S}_{\text{spec}}^{p}$ and $FC^{p}$ for the Pt6 samples in S5A and S5B Fig). Note that, using Eq 9 in Eq 11, $\mathcal{S}_{\text{spec}}^{p}(\boldsymbol{\sigma})$ reduces to $\log P_{21}^{p}(\boldsymbol{\sigma}) - \max_{p'} \log P_{21}^{p'}(\boldsymbol{\sigma})$, as we compare repertoires from the same patient and the background $P_0$ in Eq 9 is only patient, and not antigen, dependent. Antigen-stimulation experiments can still be compared two by two as in Eq 10, allowing us to estimate matrices of pairwise AUROC, see S5(C) Fig.

One can adjust a threshold above which the specificity score $\mathcal{S}_{\text{spec}}^{p}(\boldsymbol{\sigma})$ recovers clones specifically expanded under the $p$ stimulation. By varying the discrimination threshold, we build a receiver operating characteristic curve (ROC curve) to test the diagnostic ability of the score $\mathcal{S}_{\text{spec}}^{p}(\boldsymbol{\sigma})$ to detect the specificity of T-cell responses to $p$. An Area Under the ROC (AUROC) > 0.5 indicates a good classification performance. We expect such classification performance to deteriorate not only when the model is poor at capturing information about expansion, but also when there is no clear signal of specific expansion. In this case both the specificity assignment based on fold change and on the specificity score are dominated by fluctuations in counts and the AUROC becomes close to 0.5. Hence, $\mathcal{S}_{\text{spec}}^{p}(\boldsymbol{\sigma})$ can be seen as a repertoire-wide indicator itself of specificity, across the different stimulations, of T-cell responses.

## Model validation

To validate model predictions, we divide randomly every count-weighted sample into a training set (80%) and a testing set (20%). Model performance at capturing response and its specificity (through the response score in Eq 9 and the specificity score in Eq 11) is evaluated on the testing set by measuring the correlation of response scores to clone fold change (Fig 3) and by the AUROC metric of specificity (Fig 4). In this type of validation, sequences in the training and testing set are not distinct *per se*, but the counts reflecting expansion for particular sequences are different in the two sets. In short, the reads are randomly distributed between the training and testing sets. In the 5-fold leave-one-out validation protocol (S9 Fig), we instead consider samples of unique CDR3 clones and divide them into 5 sets. We use 4 sets for training models (weighting sequences by counts) and 1 set to validate the model's performance. We repeat the model training for the 5 possible training/testing partitions and we consider the average model's performance over these 5 trainings. In this protocol the same sequence cannot be found in different sets. The same 5-fold leave-one-out validation protocol was applied also for training and testing RBM models built from pools of responding-only clones, see Fig 7B–7E. At each training repetition, the testing set of responding CDR3s was mixed with generic CDR3 (randomly drawn from day 0 samples of the other patients) in the proportion 99:1 and the AUROC of discrimination between these two sets was estimated (Fig 7B and 7C reports the results for one repetition for the M1-specific repertoire from Ref. [17]).

The final AUROC (Fig 7D and 7E) were taken as the average AUROC over the 5 training repetitions.

## Repertoire dissimilarity index

We consider a measure of repertoire sequence dissimilarity that generalizes the Simpson's diversity [25, 26] (based on the probability of drawing the same clone in two independent repertoire samples) to include clone similarity instead of identity only. Following Dash et al. [17], such a measure can be defined by weighting each pairwise clone comparison by a smooth, Gaussian-like factor of the inter-clone distance. We consider the pools of only expanded clones for each assay and we look at all possible, ordered, pairwise comparisons between clones of the same pool to estimate:

$$f = \frac{1}{T} \sum_{i<j} e^{-\left(\frac{d(\boldsymbol{\sigma}_i, \boldsymbol{\sigma}_j)}{\delta}\right)^2}, \tag{12}$$

where $T$ is the total number of terms in the sum ($T = M(M-1)/2$, with $M$ number of sequences in the pool considered). $d(\boldsymbol{\sigma}_i, \boldsymbol{\sigma}_j)$ is the distance between CDR3 sequences $\boldsymbol{\sigma}_i$ and $\boldsymbol{\sigma}_j$: here we take the Levenshtein distance [37], while the analogous measure by Ref. [17] relies on the distance metric they introduced (TCRdist) calculated on the full TCR. The parameter $\delta$ sets the typical scale of the inter-sequence distance $d$ in an epitope-specific repertoire. We choose $\delta = 5.7$, which is the average CDR3 Levenshtein distance within the M1-specific repertoire from Ref. [17]. As the function $f$ measures the overall similarity within a repertoire, we define the sequence dissimilarity index as $1/f$, see Fig 7A.

## Supporting information

**S1 Fig. CDR3 length distribution.** CDR3 length distribution in the full dataset of 276993 sequences from Ref. [16].
(PDF)

**S2 Fig. Model parameters inferred by the RBM-LR model.** Same representation as in Fig 2A where, by starting with CDR3s in the *Left+Right* encoding, 2 sets of RBM weights, $w^L$ and $w^R$, are inferred. We selected for illustration $w^L_{21}$ and $w^R_6$ as they capture the sequence motifs of responding clones highlighted in Fig 5A (dark red boxes).
(PDF)

**S3 Fig. Model predicts TRB clone abundances.** A: Correlation coefficient of the RBM likelihood $\log P^p_{21}$ with clonal abundances at day 21 post-stimulation in all samples from [16]. B: The correlation coefficient shown is obtained by progressively filtering out the low-abundance sequences from the testing set, as illustrated for the Pt3 NA dataset. C: Scatter plot between clone abundance at day 21 and the RBM likelihood $\log P^{NA}_{21}$ for the full Pt3 NA dataset, showing that the correlation is poor for clones at low counts. The golden dashed line marks the minimal clone abundance that is considered to measure the correlation coefficient reported in A and indicated by the golden dot in B. B-D contain the comparison between the performance of the RBM and SONIA, confirming that capturing sequence correlations (as the RBM does, see also S4 Fig) ensures better prediction of clonal abundance.
(PDF)

**S4 Fig. Model selection.** A: Hyperparametric search for training SONIA on count-weighted datasets: we set the $L_2$ regularization at the intermediate value 0.01 (black dot), which ensures good performance at capturing both clone abundance (through $\log P^p_{21}$) and clonal fold

change (through $\mathcal{S}_{\text{resp}}^{p}$) in the validation set. B. Data statistics (single-site frequency and connected correlations) are reproduced by SONIA. SONIA is documented to perform best when trained on lists of unique sequences [15], where also correlations are well reproduced (see inset). As a comparison, the gray points stand for the marginals given by the generation model $P_{\text{gen}}$. C-D: Hyperparametric search for training RBM on count-weighted datasets. Similarly to A, it is performed based on how clonal abundances and clone fold change are reproduced by $\log P_{21}^{p}$ and $\mathcal{S}_{\text{resp}}^{p}$. We identify as optimal the combinations of parameters indicated by the black box, which result in maximal correlations with fold change (D) while keeping the correlation to abundance in the intermediate-high range (C). F-G: Hyperparametric search for training RBM on datasets of only responding CDR3s. Optimal $N^{h}$ and $\lambda_{1}^{2}$ (black dot) are chosen to ensure high likelihood $\mathcal{L}$ on the training set and to prevent overfitting on the validation set. E-H show that in both cases single-site amino acid frequencies and connected correlations are extremely well reproduced by the RBM (with parameters marked by the black dot in D and G). In all hyperparametric searches shown, the model training set consist of 80% of all sequences in the dataset, the validation set is the remaining 20%. Datasets used: Pt3 NA from [16] (A-E), samples from Ref. [17] (F-H).
(PNG)

**S5 Fig. RBM prediction of response specificity.** A-B: Differential degree of clone expansion under stimulation $p$ is mapped by the model into differential response scores $\mathcal{S}_{\text{resp}}^{p}$, enabling a model-based assessment of response specificity. Here we consider sequences from sample Pt6 NA1, where clone abundance reflects response to NA1. The fold change due to NA1 stimulation, $FC^{NA1}$, is on average higher than the fold change measured for all the other antigens $p$ tested in the same patient, $FC^{p}$ (A), suggesting a specific response. This set of NA1-specific responders is on average assigned higher response score by the RBM model trained on the Pt6 NA1 dataset, $S_{\text{resp}}^{NA1}$, than by models trained on the other Pt6 samples, $S_{\text{resp}}^{p}$, where the same sequences behave as unspecific responders (B). All models are trained on 80% of a given sample, and we show average log fold change $\log FC^{p}$ and response scores $\mathcal{S}_{\text{resp}}^{p}$ (both expressed as log base 10) over the Pt6 NA1 testing set. C: Matrices of RBM pairwise AUROC for Pt1, Pt2, Pt4 and Pt6 (patients for whom more than 2 antigens were tested, see Table 1). The value at each row/column intersection gives the AUROC (estimated through RBM scores) of response specificity between the antigen to which the row and column refer.
(PNG)

**S6 Fig. Performance comparison to Position Weight Matrix (PWM).** PWM is the simplest sequence-based modeling strategy and it is here learnt on aligned count-weighted datasets. The PWM probability assigned to each CDR3 clone $\boldsymbol{\sigma}$ is factorized over CDR3 positions, *i.e.* $P_{t}^{p}(\boldsymbol{\sigma}) = \prod_{i=1}^{N^{\sigma}} P_{t,i}^{p}(\sigma_{i})$ and $P_{t,i}^{p}(\sigma_{i})$ is taken as the frequency of amino acid $\sigma_{i}$ at position $i$ in the $p$-specific repertoire at time $t = 0, 21$ (days). PWM performance in terms of correlation between response score $\mathcal{S}_{\text{resp}}^{p}$ and clone fold change (see Fig 3) and AUROC of specificity (see Fig 4) is compared to SONIA (A,C) and RBM (B,D) for all samples from [16].
(PDF)

**S7 Fig. RBM recovers sequence motifs analyzed in Dash et al. 2017 [17] and Glanville et al. 2017 [22].** A: One of the set of weights ($w_4$) learned by the RBM (same model as in Fig 6B) picks up the pp65$_{495}$ motif discussed in [17]. B: The clones characterized by this motif can be identified by the negative values of the projection onto this weight (input to $\mathbf{h}_4$), indicated by the black arrow. C-D: Same representation as in A-B for the RBM trained on EBV, CMV, Flu data from [22] (same model as in Fig 6D), where one weight highlights the sequence motif

found in [22] for the pp65-specific clones. E: Sets of weights of a RBM trained on the CDR3 sequences of *M. tuberculosis*-specific CD4+ T cells from [22] recover three motifs from the five representative TCR specificity groups analyzed in [22]. The set of weights $w_4$ picks up a gap motif (symbols ⊟) indicating that the motif 'GGE' is found in the center of longer CDR3 sequences, where there are gap insertions in the other sequences. F: We choose for illustration two specificity groups: their sequences (listed in the boxes) are characterized by high values of the projection onto the corresponding weights (here the inputs to $h_4$ and $h_{13}$).
(PDF)

**S8 Fig. RBM dimensionality reduction is consistent with GLIPH clusters enriched in clonally expanded T cells.** A-B: Clustering by GLIPH version 2 (GLIPH2 [23]) of the TRB sequences from respectively the datasets Pt3 NA and Pt5 CR (same datasets considered in Fig 5). We plot for each cluster the "Fisher score" reported by GLIPH2 (a p-value quantifying the significance of the motif characterizing the cluster compared to a reference repertoire) and the "expansion score" (a p-value quantifying the significance of the abundance of clones in the cluster) and we annotate the most significant clusters under both criteria by the corresponding sequence motif. GLIPH2 builds several clusters at high expansion score precisely around the motifs characteristic of expanded clones identified by RBM, as we make visible in the amino acid letters marked in color in A-B and C-D (here the same plots as in Fig 5 are reported). Clusters at highest motif-wise significance contain unexpanded clones, hence their motifs do not appear in the RBM dimensionality reduction C-D, where we kept only clones with top RBM scores (expanded). All the sequences not marked in color in C-D are expanded but not clustered by GLIPH2; in A-C, other expanded clones falling into a GLIPH2 cluster are indicated by small red arrows. We run GLIPH2 using the web tool available at http://50.255.35. 37:8080/ with options: *Reference version = version 2.0, Reference = CD8, all aa interchangeable = YES* and providing as input for each dataset the list of CDR3 sequences, V and J segments and sequence counts at day 21. The clusters identified by GLIPH2 are 1113 for the Pt3 NA sample (total number of sequences = 14228) and 1759 for the Pt5 CR sample (total number of sequences = 14122).
(PDF)

**S9 Fig. Sequence dissimilarity among responding clones and model generalization.** A: The *x*-axis gives the correlation, averaged over trainings, between the RBM score $\log P^p_{21}$ and clonal abundance 21 days post-stimulation in a 5-fold leave-one-out validation protocol (see Materials and methods) for all the Pt1,...,Pt7 datasets from [16]. The scatter plot shows that the ability of the RBM score to recover clonal abundance increases for low diversity repertoires (Pearson correlation *r* of magnitude |*r*| = 0.61, *p*-value for testing non-correlation = 0.002). B: Correlation between the RBM $\log P^p_{21}$ and clone abundance for the dataset at lowest dissimilarity index (Pt4 WT), both for training and testing set, as a function of the dataset size (filtered by counts). A correlation in the testing set higher than zero is recovered only when retaining the most abundant clones: the dataset size chosen for the points in A,C—indicated by the dark red dot—corresponds to $\sim$ 160 sequences. The trend, markedly different for training and testing set, signals overfitting that is unavoidable when the response is heterogeneous (as quantified by the sequence dissimilarity index, Fig 7A). C: Correlation to clone abundance by RBM—same quantity as in A—is compared to the one obtained by SONIA.
(PDF)

**S1 Data. Numerical data used in Figs 3 and 8.**
(XLSX)

**S2 Data. Numerical data used in Figs 4 and 8.**
(XLSX)

**S3 Data. Numerical data used in Fig 7.**
(XLSX)

## Acknowledgments

We acknowledge helpful exchanges with Jorge Cossio-Diaz, Thomas Dupic, Yuval Elhanati, Giulio Isacchini, Marta Łuksza, Cosimo Lupo, Carlos Rodriguez, Luis A. Rojas, Zachary Sethna, Alexander Solovyov, Jérôme Tubiana. S.C., B.D.G. and R.M. thank the Aspen Center for Physics for their ospitality during the initial part of this work.

## Author Contributions

**Conceptualization:** Barbara Bravi, Vinod P. Balachandran, Benjamin D. Greenbaum, Aleksandra M. Walczak, Thierry Mora, Rémi Monasson, Simona Cocco.

**Data curation:** Barbara Bravi.

**Formal analysis:** Barbara Bravi, Aleksandra M. Walczak, Thierry Mora, Rémi Monasson, Simona Cocco.

**Funding acquisition:** Vinod P. Balachandran, Benjamin D. Greenbaum, Aleksandra M. Walczak, Thierry Mora, Rémi Monasson, Simona Cocco.

**Investigation:** Barbara Bravi, Aleksandra M. Walczak, Thierry Mora, Rémi Monasson, Simona Cocco.

**Methodology:** Barbara Bravi, Aleksandra M. Walczak, Thierry Mora, Rémi Monasson, Simona Cocco.

**Software:** Barbara Bravi.

**Supervision:** Aleksandra M. Walczak, Thierry Mora, Rémi Monasson, Simona Cocco.

**Validation:** Barbara Bravi, Aleksandra M. Walczak, Thierry Mora, Rémi Monasson, Simona Cocco.

**Visualization:** Barbara Bravi.

**Writing – original draft:** Barbara Bravi, Vinod P. Balachandran, Benjamin D. Greenbaum, Aleksandra M. Walczak, Thierry Mora, Rémi Monasson, Simona Cocco.

**Writing – review & editing:** Barbara Bravi, Vinod P. Balachandran, Benjamin D. Greenbaum, Aleksandra M. Walczak, Thierry Mora, Rémi Monasson, Simona Cocco.

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
