## [Decision Letter · Decision Letter 0]

7 May 2021

Dear Dr Cocco,

Thank you very much for submitting your manuscript "Probing T-cell response by sequence-based probabilistic modeling" for consideration at PLOS Computational Biology. As with all papers reviewed by the journal, your manuscript was reviewed by members of the editorial board and by several independent reviewers. The reviewers appreciated the attention to an important topic. Based on the reviews, we are likely to accept this manuscript for publication, providing that you modify the manuscript according to the review recommendations.

Both reviewers liked the paper. As pointed out by both reviewers and reviewer 2 in particular making it more accessible to math biologists not expert in machine learning might greatly enhance its impact. I know this could take quite a lot of work and time but I think it would be worth the effort.

Sincerely,

Rustom Antia

Associate Editor

PLOS Computational Biology

Rob De Boer

Deputy Editor

PLOS Computational Biology

[LINK]

Both reviewers liked the paper. As pointed out by both reviewers and reviewer 2 in particular making it more accessible to math biologists not expert in machine learning might greatly enhance its impact. I know this could take quite a lot of work and time but I think it would be worth the effort. So while a minor revision would make it acceptable for publication ...

Reviewer's Responses to Questions

**Comments to the Authors:**

Reviewer #1: In this work the Authors present a family of new probabilistic strategies to

analyze T cell receptor (TCR) repertoire. The underlying idea is to construct

probabilistic models on the base of clone abundances to extract TCR sequence

motifs.

The Authors present 3 methods: Restricted Boltzmann Machines (RBM) on previously

aligned TCR sequences, a selection-factor based model (SONIA), and a new RPM

strategy that does not require previous alignment of the sequences. The

application ground of the method consists of TRB CDR3 regions sequences of

7 patient repertoire sequences taken at different times (21 days apart).

The manuscript is scientifically sound and I find particularly interesting the

fact that overall the RBM results used on the not-aligned dataset provides

analogous results compared to the RBM on aligned sequences and SONIA. Moreover,

the lower dimensional "projection" of the RBM on the latent variable space

present sequence motives that were previously found in other works (notably Dash

et al. [17], and Glanville et al [21]).

The manuscript is well written (apart from some issues that I will report

below), and it is scientifically sound. Previous work in the field has been

fairly taken into account in the bibliography.

In the following I will present a series of minor comments that the Author should

address for the sake of clarity.

1. Datasets: I personally find the description of Balachandran et al [16]

dataset in Section Results - Dataset Structure a bit too unforgiving for those

computational biologists which do not have a solid background in immunology. In

particular, without referring to the original paper, it was not clear to me the

preliminary discussion on the neoantigen model. The first paragraph (lines

79-90) is really dense, and would benefit from a critical rewriting.

2. Presentation of the results: Although the discussion is nice and thorough I

found a bit hard to figure out eventually which one of the three strategies was

better suited for this kind of studies. Perhaps a final table (or another

histogram) comparing the global and per patient AUROC for the 3 methods,

expanding somehow Fig.3 D would be helpful.

3. By construction (iterative alignment strategy) the structure of the aligned

sequences have gaps in the center and letter at the extremities. Could the LR

strategies be used (I do not know how) to prove or disprove this structure?

Reviewer #2: Bravi et al. proposes to use a machine-learning approach known as Restricted Boltzmann Machine (RBM) to identify antigen-specific TCRs with a sequence-based inference approach. Several improvements and clarifications are needed so that biologists/immunologists (and people outside the field of machine learning) could appreciate the results of the paper.

1. Figure 1 shows the schematic of the approach. I found Figure 1A and its description in the text to be confusing as it is missing some important immunological details and may be misleading. First, not only antigen but also cytokines should be added to maintain the T cells during 21 days. Second, peptides decay quickly, so T cells probably were restimulated several times between day 0 and day 21, and it should be indicated on the schematic. Third, you need to say how long the cells were stimulated at day 21 before they were taken for analysis.

2. It looks like the patterns (“constraints”) for antigen recognition are very specific for each person/antigen. Do you need the individual person/antigen data for PBMCs expansion in vitro to obtain these patterns for a new patient or already analyzed patients and single time point data from a new patient would be sufficient? It is not clear from the description of the results how it could be generalized. For example, how much can be inferred from a single patient’s blood draw (analog of PBMCs at day 0 in Figure 1A)? Will it be possible to infer the number of neoantigens (important prognostic factor) from a single blood draw? What determines the percentage of well-clustered and expanded sequences in different individuals? Please, describe the practical use of the results for the biologists/immunologists more clearly in the main text, discussion, and abstract.

3. Please describe the limitations: what the method could and could not do in terms of answering the related biological questions.

4. The last sentence in the abstract is unclear. I suggest replacing it with a more specific example(s) of practical use of the RBM approach (see point 2).

**Have all data underlying the figures and results presented in the manuscript been provided?**

Reviewer #1: Yes

PLOS authors have the option to publish the peer review history of their article (what does this mean?). If published, this will include your full peer review and any attached files.

Reviewer #1: No

Reviewer #2: No

**Have the authors made all data and (if applicable) computational code underlying the findings in their manuscript fully available?**

Reviewer #2: **No: **no code

Figure Files:

Data Requirements:

Reproducibility:

References:

---

## [Editor Report · Decision Letter 1]

22 Jul 2021

Dear Dr Cocco,

We are pleased to inform you that your manuscript 'Probing T-cell response by sequence-based probabilistic modeling' has been provisionally accepted for publication in PLOS Computational Biology.

Best regards,

Rustom Antia

Associate Editor

PLOS Computational Biology

Rob De Boer

Deputy Editor

PLOS Computational Biology

---

## [Editor Report · Acceptance letter]

27 Aug 2021

PCOMPBIOL-D-20-02319R1 

Probing T-cell response by sequence-based probabilistic modeling

Dear Dr Cocco,

I am pleased to inform you that your manuscript has been formally accepted for publication in PLOS Computational Biology. Your manuscript is now with our production department and you will be notified of the publication date in due course.

With kind regards,

Livia Horvath
